# Restricted cell cycle is essential for clonal evolution and therapeutic resistance of pre-leukemic stem cells

Cedric S. Tremblay [1], Jesslyn Saw[1], Sung Kai Chiu[1,2], Nicholas C. Wong[3], Kirill Tsyganov[3], Sarah Ghotb[1], Alison N. Graham[4], Feng Yan[1,3], Andrew A. Guirguis [1,2], Stefan E. Sonderegger[1], Nicole Lee[1], Paul Kalitsis [4], John Reynolds [5], Stephen B. Ting[1,2], David R. Powell [3,6], Stephen M. Jane[2,7] & David J. Curtis[1,2]

Pre-leukemic stem cells (pre-LSCs) give rise to leukemic stem cells through acquisition of additional gene mutations and are an important source of relapse following chemotherapy. We postulated that cell-cycle kinetics of pre-LSCs may be an important determinant of clonal evolution and therapeutic resistance. Using a doxycycline-inducible *H2B-GFP* transgene in a mouse model of T-cell acute lymphoblastic leukemia to study cell cycle in vivo, we show that self-renewal, clonal evolution and therapeutic resistance are limited to a rare population of pre-LSCs with restricted cell cycle. We show that proliferative pre-LSCs are unable to return to a cell cycle-restricted state. Cell cycle-restricted pre-LSCs have activation of p53 and its downstream cell-cycle inhibitor p21. Furthermore, absence of p21 leads to proliferation of pre-LSCs, with clonal extinction through loss of asymmetric cell division and terminal differentiation. Thus, inducing proliferation of pre-LSCs represents a promising strategy to increase cure rates for acute leukemia.

[1] Australian Centre for Blood Diseases, Central Clinical School, Monash University, Melbourne, VIC 3004, Australia. [2] Department of Clinical Haematology, Alfred Hospital, Prahran, VIC 3181, Australia. [3] Monash Bioinformatics Platform, Monash University, Clayton, VIC 3800, Australia. [4] Murdoch Children's Research Institute, Royal Children's Hospital, Parkville, VIC 3052, Australia. [5] Biostatistics Consulting Platform, Monash University, Clayton, VIC 3800, Australia. [6] School of Computer Science and Software Engineering, Monash University, Clayton, VIC 3800, Australia. [7] Department of Medicine, Central Clinical School, Monash University, Melbourne, VIC 3004, Australia. Correspondence and requests for materials should be addressed to D.J.C. (email: david.curtis@monash.edu)

The leukemia stem cell (LSC) concept posits the presence of a cell population with stem cell-like properties enabling their ability to generate the full heterogeneity of the tumor and fuel tumor growth during disease progression. These LSCs are intrinsically resistant to therapies via potential mechanisms that include quiescence, low reactive oxygen stress, enhanced DNA repair and expression of adenosine triphosphate-binding cassette transporters. Over recent years, genome-wide studies of matched primary and relapsed leukemic samples strongly support this model wherein the clone responsible for relapse arises from either a pre-existing LSC or an antecedent LSC clone referred to as a pre-leukemic stem cell (pre-LSC)[1–3]. These pre-LSCs contain the founding genetic mutation but not the full complement of mutations found at diagnosis. Although pre-LSCs retain the ability to differentiate into functional mature blood cells, they also have long-lived self-renewal capacity[4] and their presence in patient remission samples following intensive chemotherapy portends a high risk of relapse[5]. In addition to acute leukemia, cells akin to pre-LSCs underpin myelodysplastic syndromes and perhaps even clonal hematopoiesis of the elderly, which can evolve into acute leukemia over many months to years[6,7].

Quiescence may be an important mechanism of therapeutic resistance for LSCs, particularly for therapies that rely upon cell proliferation for their activity. Clinically, this concept is exemplified in chronic myeloid leukemia where, even in the era of tyrosine kinase inhibitor therapy, the absence of cure is thought to reside with the inability to eradicate the quiescent clones of LSCs[8–10]. Perhaps the most convincing in vivo evidence comes from Ebinger et al.[11], who identified a rare subpopulation of dormant and treatment-resistant cells in patient-derived xenografts. They also showed that these chemoresistant cells share the same gene expression profile with primary leukemia cells isolated from patients at minimal residual disease. Moreover, Saito et al.[12] experimentally showed that quiescent leukemic cells residing in the bone marrow niche were protected from chemotherapy. They subsequently showed that overcoming quiescence with cytokine stimulation could sensitize these leukemogenic cells to chemotherapy. However, these and other experimental in vivo studies of LSC quiescence have almost exclusively used label-retaining cell fixation assays with DNA analogs such as bromodeoxyuridine which preclude subsequent functional studies[13]. This major hurdle for the study of quiescence in hematopoietic stem and progenitor cells has recently been overcome by the generation of transgenic mice expressing a doxycycline-regulated histone H2B-GFP fusion product that is incorporated into the nucleosome during cell division[14,15]. Prospective isolation of quiescent hematopoietic stem cells (HSCs) based on cell surface markers and green fluorescent protein (GFP) retention showed that quiescent HSCs are both enriched for long-term repopulating activity and the source of proliferative HSCs during times of stress. To our knowledge, these H2B-GFP mice have been reported only once in the leukemia context. In this study, oncogenic RAS induced a bimodal effect on HSC cycling, with the quiescent but not proliferative fraction outcompeting healthy HSCs[16]. However, the relationship between quiescence and chemoresistance or clonal evolution remained to be explored.

Aberrant expression of *LMO2* through chromosomal translocation or a somatically acquired neomorphic promoter occurs in 50% of T-cell acute lymphoblastic leukemia (T-ALL)[17,18]. Using a mouse model of T-ALL driven by the *Lmo2* oncogene, we reported the identification of cells that fulfill the fundamental properties of pre-LSCs, namely self-renewal potential without a block in differentiation[19]. Transplant studies showed that pre-LSCs arise from immature $CD4^-CD8^-CD25^+CD44^-$ (DN3) T-cell progenitors in *Lmo2*-transgenic mice (*Lmo2*Tg), as these cells were capable of long-term repopulation capacity in recipient

mice. These self-renewing DN3 cells retained T-cell differentiation potential but eventually gave rise to T-ALL as they accumulated additional lesions that promote leukemia progression[20,21]. Importantly, these pre-LSCs could survive and recover after high-dose radiation[19]. Here, we have used the doxycycline-inducible H2B-GFP mouse model crossed with the *Lmo2*-transgenic mice to study the importance of cell cycle in pre-LSCs. We show that self-renewal, clonal evolution and therapeutic resistance are limited to a rare population of pre-LSCs with restricted cell cycle. Importantly, proliferative pre-LSCs are unable to return to a cell cycle-restricted state. Thus, inducing proliferation of pre-LSCs represents a promising strategy to increase cure rates for acute leukemia.

## Results

**Identification of cell cycle-restricted pre-LSCs.** We crossed the *TetOP-H2B-GFP*KI/+ mouse line with *Lmo2*Tg mice to examine the cell-cycle kinetics of pre-LSCs. Heterozygous *TetOP-H2B-GFP*KI/+;*Lmo2*Tg (H2B-GFP;*Lmo2*Tg) mice were treated with doxycycline for 6 weeks to induce expression of H2B-GFP in dividing cells. We then examined GFP expression in thymocytes following withdrawal of doxycycline for 1, 2, 4 and 8 weeks (Fig. 1a), focusing on the DN3 T-cell fraction, which contains all pre-LSC activity[19,22]. At the end of the labeling period, almost all DN3 cells in both control and *Lmo2*Tg mice expressed the H2B-GFP division marker, which comprised high and intermediate populations (Fig. 1b). Interestingly, a small proportion of DN3 cells in H2B-GFP;*Lmo2*Tg mice remained GFP negative despite a 6-week labeling period, which suggests the presence of cells that had not divided. Consistent with this highly proliferative stage of T-cell development, withdrawal of doxycycline led to a rapid loss of GFP in DN3 thymocytes from control H2B-GFP mice such that there were no GFPhi cells beyond 2 weeks. However, in H2B-GFP;*Lmo2*Tg mice, a small fraction of DN3 cells retained GFP expression for up to 8 weeks (Fig. 1b). In absolute numbers, this rare GFPhi population of cells at 8 weeks equated to 3000 cells per whole thymus (Supplementary Fig. 1a). Consistent with the exclusive presence of pre-LSCs within the DN3 thymocyte population, no difference was observed in the proportion of GFP-retaining cells within other T-cell subsets in H2B-GFP;*Lmo2*Tg mice compared with control mice (Supplementary Fig. 1b). Assuming that the mean cell fluorescence halved with each cell division, we estimated a mean cycling time of $18.2 \pm 2.8$ h for control DN3 cells and $50.1 \pm 13.8$ h for *Lmo2*Tg DN3. Given there was no obvious long-term plateau of GFP loss (Fig. 1c), we designated GFPhi cells as cell-cycle restricted rather than quiescent or dormant. Staining with 4′,6-diamidino-2-phenylindole (DAPI) and Ki67, an independent assay for analyzing snapshots of cell cycle, confirmed that most GFPhi cells present beyond 2-week chase were in the $G_0$ phase (Fig. 1d), whereas less than 40% of GFPlo cells were in the $G_0$ phase when assayed 2 weeks after withdrawal of doxycycline (Fig. 1e). Therefore, all subsequent analyses were performed using DN3 thymocytes from mice 2 weeks after withdrawal of doxycycline and cells that did not divide during the 6-week labeling period were excluded.

**Cell-cycle restriction maintains leukemogenic pre-LSCs.** Serial transplantation is the gold-standard assay that defines LSCs[23]. Fluorescence-activated cell sorting (FACS)-isolated GFPhi and GFPlo DN3 cells from 2-month-old H2B-GFP;*Lmo2*Tg mice were injected into sublethally irradiated $CD45.1^+$ recipient mice to examine the importance of cell-cycle kinetics on repopulating activity (Fig. 2a). In primary recipients, GFPhi DN3 cells were able to expand 100-fold compared with 10–20-fold for GFPlo DN3 cells (Fig. 2b). This decreased capacity to generate DN3 cells

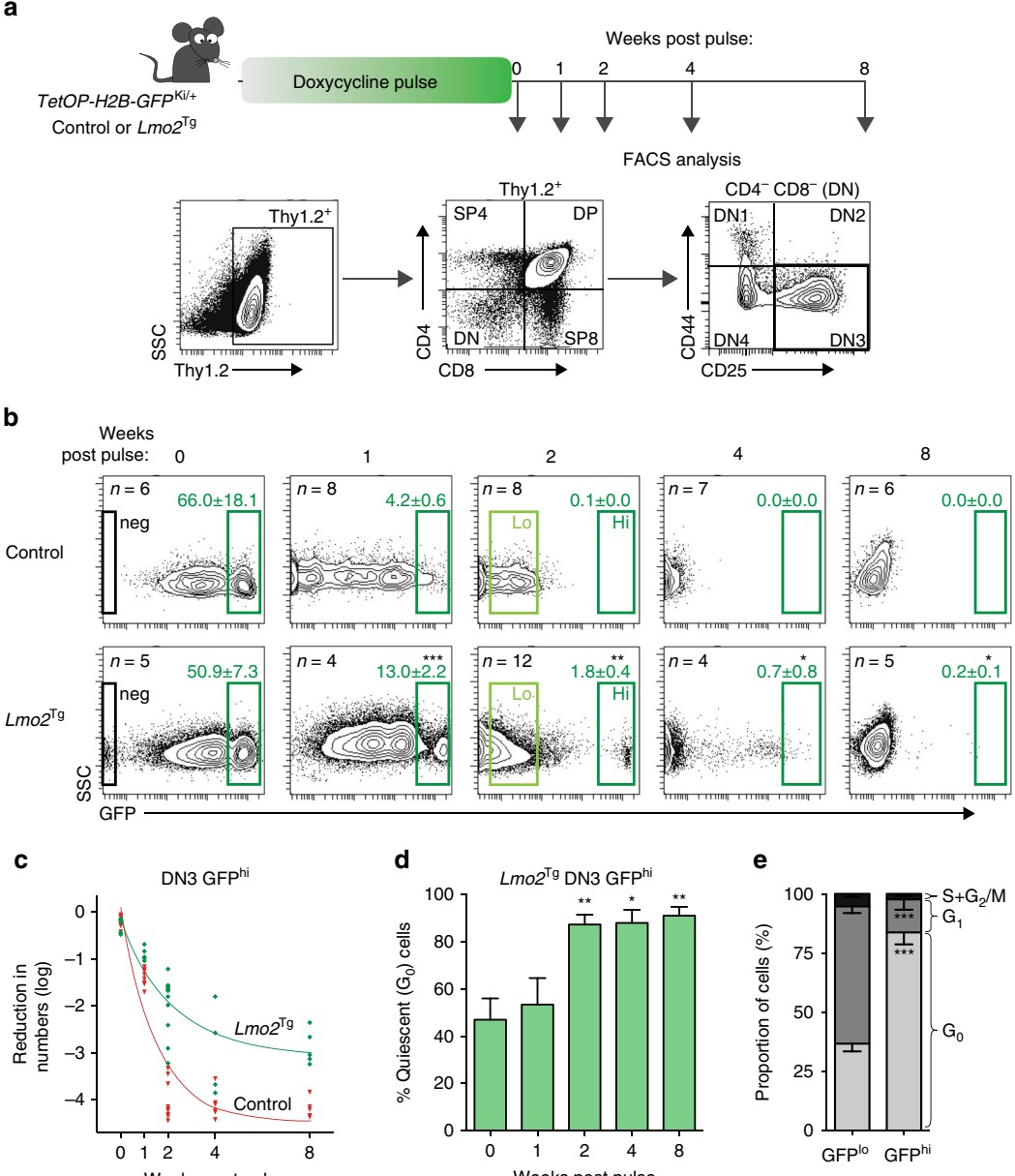

**Fig. 1** Cell-cycle kinetics of green fluorescent protein (GFP)-labeled populations of DN3 thymocytes. **a** Experimental design for the analysis of cell-cycle kinetics of DN3 thymocytes. **b** Representative flow cytometric analysis of GFP expression in DN3 thymocytes from *H2B-GFP;Lmo2*[Tg] mice and littermate controls after labeling, followed by 0, 1, 2, 4 or 8 weeks of chase without doxycycline, as indicated. Unlabeled cells (neg), GFP[lo] and GFP[hi] populations are framed, with the average proportion (mean ± s.d.) of GFP[hi] cells indicated, Student's *t*-test (vs control). **c** Kinetics of GFP-labeling loss in DN3 thymocytes from 6-week-old H2B-GFP (red) and *H2B-GFP;Lmo2*[Tg] mice (green), after the doxycycline pulse. **d** Progressive proportion of cell cycle-restricted cells within the GFP[hi] DN3 population from *H2B-GFP;Lmo2*[Tg] mice. Values are mean ± s.d., Student's *t*-test, as compared to the 0 week post-pulse time point. **e** Cell-cycle analysis in GFP[lo] and GFP[hi] DN3 cells from *H2B-GFP;Lmo2*[Tg] mice, 2 weeks after removal of doxycycline. Values are mean ± s.e.m., Student's *t*-test (vs control) *$p < 0.05$, **$p < 0.01$, ***$p < 0.001$

may be explained in part by the enhanced differentiation of GFP[lo] DN3 cells into CD4[+]CD8[+] double-positive (DP) thymocytes (Fig. 2c), which lack self-renewal activity[19,22]. We performed serial transplant to assess long-term self-renewal capacity (Fig. 2a), the quintessential property of all stem cells. A period of 4 weeks between transplants was chosen to allow competition with normal HSCs, which take up to 3 weeks to repopulate the thymus. GFP[hi] DN3 cells retained an ability to expand for at least four rounds of transplantation (Fig. 2d). In contrast, GFP[lo] DN3 cells progressively lost the ability to regenerate DN3 cells such

that by the fourth passage, there was exhaustion of their expansion potential. Thus, restricted cell cycle was a critical property of self-renewing pre-LSCs.

Given that self-renewal enables pre-LSCs to accumulate additional genetic events necessary for progression to leukemia, we postulated that only the GFP[hi] DN3 cells would be capable of generating T-ALL. Consistent with this idea, there was increased monoclonality in GFP[hi] DN3 cells as measured by *Tcrβ* rearrangement (Supplementary Fig. 2a)[24]. Furthermore, a proportion of secondary, tertiary and quaternary recipients of

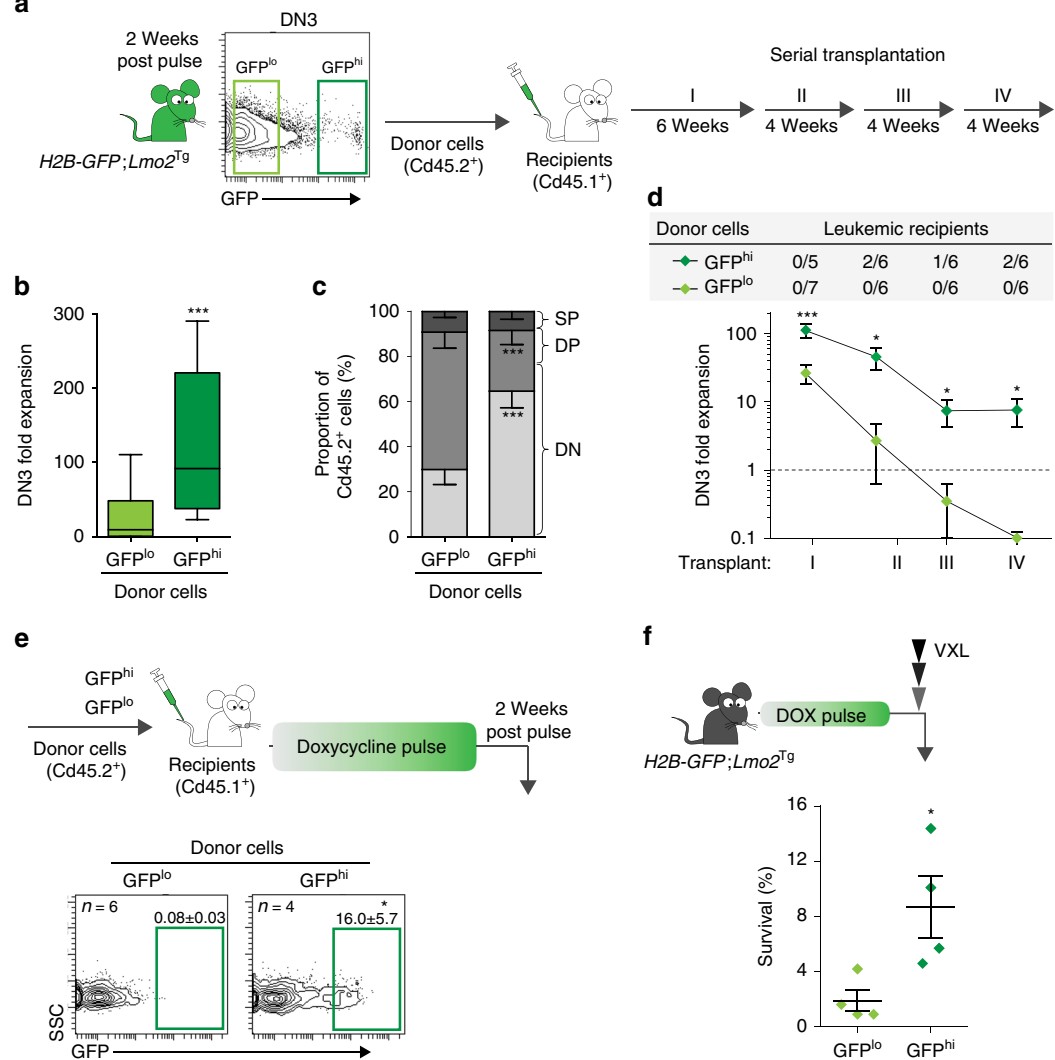

**Fig. 2** Functional characterization of cell cycle-restricted pre-leukemic stem cells (pre-LSCs). **a** Scheme for serial transplantation into primary (I), secondary (II), tertiary (III) and quaternary (IV) recipients, using purified GFP$^{lo}$ and GFP$^{hi}$ DN3 thymocytes. **b** Fold expansion of donor-derived DN3 thymocytes enumerated in the thymus of primary recipient mice. Values are median ± 95% confidence interval (CI), Student's $t$-test, $n = 15$. **c** Immunophenotype of donor-derived cells in the thymus of primary recipients. DN, DP and SP represent the CD4$^-$CD8$^-$ double-negative, CD4$^+$CD8$^+$ double-positive and CD4$^+$CD8$^-$ single-positive with CD4$^-$CD8$^+$ single-positive populations, respectively. Values are mean ± s.d., Student's $t$-test, $n = 5$. **d** Fold expansion of donor-derived DN3 thymocytes in the thymus of recipients serially injected with either GFP$^{lo}$ or GFP$^{hi}$ DN3 $Lmo2^{Tg}$ thymocytes. The frequency of recipients ultimately developing donor-derived leukemia is indicated in the top gray shade diagram. Number of leukemic recipients over the total number of recipients injected per cohort is indicated. Values are mean ± s.e.m., Student's $t$-test. **e** Scheme for green fluorescent protein (GFP) labeling of donor-derived DN3 thymocytes in primary transplants (top) and proportion of donor-derived GFP$^{hi}$ DN3 cells (bottom, framed) in the thymus of primary recipients, 2 weeks post pulse. Values are mean ± s.d., Student's $t$-test. **f** Proportion of chemoresistant DN3 thymocytes from each GFP-labeled subset found in *H2B-GFP; Lmo2*$^{Tg}$ mice 2 weeks after doxycycline pulse. VXL induction-like therapy for T-ALL, including vincristine, dexamethasone and ʟ-asparaginase[27]. Values are mean ± s.d., Student's $t$-test *$p < 0.05$, ***$p < 0.001$

GFP$^{hi}$ DN3 cells developed T-ALL, whereas no cases of leukemia were observed in mice transplanted with proliferative GFP$^{lo}$ DN3 cells over the 22-week serial transplant period (Fig. 2d and Supplementary Fig. 2b). Given that leukemias only arise in recipients injected with GFP$^{hi}$ cells, our results demonstrate that restricted cell cycle is important for clonal evolution and leukemogenic potential of pre-LSCs.

HSCs can re-enter a dormant state following hematopoietic stress, including chemotherapy[25,26]. To determine if proliferative pre-LSCs were able to return to a cell cycle-restricted state, we administered doxycycline for 6 weeks to mice transplanted with GFP$^{lo}$ cells. Unlike normal HSCs, proliferative pre-LSCs were unable to generate GFP$^{hi}$ cells (Fig. 2e and S2c). In contrast, GFP$^{hi}$ cells generated a cell cycle-restricted progeny, confirming that they can be maintained even in the setting of proliferative stress. To determine whether cell cycle-restricted pre-LSCs are maintained in disease progression, we determined the numbers of GFP$^{hi}$ DN3 cells in 6-month-old *Lmo2*$^{Tg}$ mice. In these older mice, the GFP$^{hi}$ DN3 cell population had expanded 3-fold, and 10-fold in mice with overt T-ALL (Supplementary Fig. 2d). Thus, cell cycle-restricted pre-LSCs expand with disease progression.

A recent study by Ebinger et al.[11] linked cell-cycle restriction with treatment resistance in human ALL. To determine whether cell-cycle restriction protects pre-LSCs against chemotherapeutic

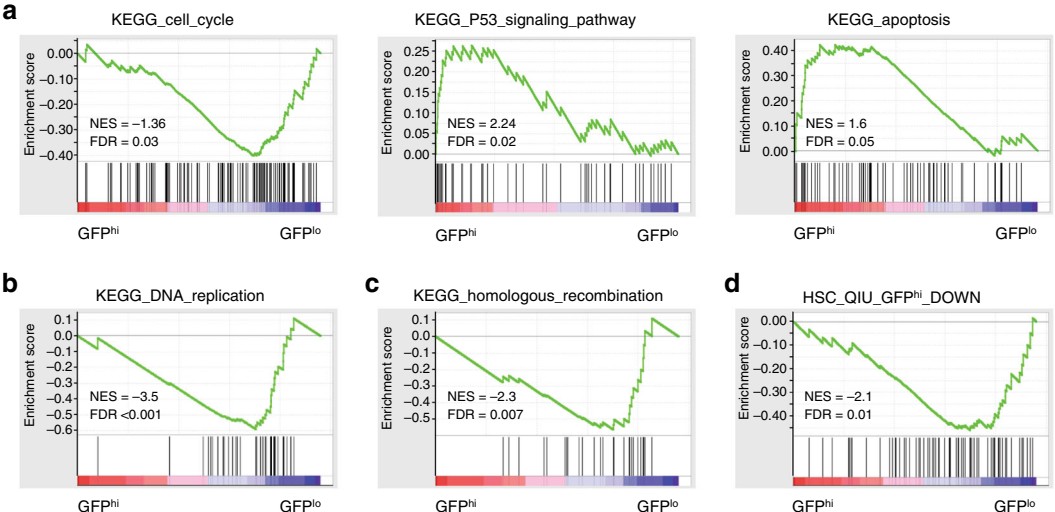

**Fig. 3** Gene expression profiling of cell cycle-restricted pre-leukemic stem cells (pre-LSCs). Gene set enrichment analysis (GSEA) of **a** cell cycle, p53-mediated response and apoptosis genes, **b** DNA replication genes, **c** homologous recombination genes and **d** cell-cycle genes downregulated in quiescent GFP[hi] HSCs from ref.[29]. GFP green fluorescent protein, FDR false discovery rate, NES normalized enrichment score in GFP[hi] as compared to GFP[lo] population

agents typically used for human T-ALL, we looked for enrichment of GFP[hi] DN3 cells following treatment of *H2B-GFP;Lmo2*[Tg] mice with a combination of vincristine, dexamethasone and L-asparaginase (VXL)[27]. Consistent with the cell cycle-dependent effect of chemotherapy, the proportion of cells surviving 24 h after combination therapy was fourfold higher in the GFP[hi] fraction compared with the GFP[lo] fraction: 9% of GFP[hi] cells compared with 2% of GFP[lo] cells (Fig. 2f). A similar increased resistance of GFP[hi] cells was observed in response to sub-lethal irradiation (Supplementary Fig. 2e). Given that the H2B-GFP labeling model reflects the history of the cell cycle, the enrichment for GFP[hi] cells observed cannot be due to therapy-induced senescence and must reflect cells that have not actively divided in the preceding 2 weeks of chase. Thus, cell-cycle restriction of pre-LSCs enhances resistance to therapeutic agents used for treatment of human T-ALL.

**Expression profile of cell cycle-restricted pre-LSCs.** To investigate the molecular signature of cell cycle-restricted pre-LSCs, we performed gene expression profiling of GFP[hi] and GFP[lo] DN3 cells from three independent cohorts of *H2B-GFP;Lmo2*[Tg] mice. Overall, there were 853 genes differentially expressed more than twofold using a false discovery rate (FDR) of 0.01: 255 genes increased and 598 genes reduced in the GFP[hi] cells (Supplementary Fig. 3a and Supplementary Data 1). Gene ontology analysis of upregulated genes showed enrichment for p53 activation (*Cdkn1a, Sesn2, Ddit4, Pmaip1, Zmat3* and *Phlda3*), antigen processing and presentation (*H2-DMb2, CD74, H2-Aa, H2-Eb1, H2-Q1, H2-T3* and *H2-Q6*) and T-cell differentiation (Supplementary Data 1). Gene set enrichment analysis (GSEA) confirmed downregulation of cell-cycle regulators, as well as the activation of p53 apoptosis and the antigen processing pathway (Fig. 3a and Supplementary Data 1) and also identified increased expression of multiple proteasomal activator subunits (Supplementary Fig. 3b and Supplementary Data 1). GSEA also identified reduced expression of multiple DNA polymerase and mini-chromosome maintenance (MCM) genes required for DNA replication (Fig. 3b and Supplementary Data 1). Consistent with the reported error-prone DNA repair in quiescent HSCs[28], expression of genes required for high-fidelity homologous

recombination and mismatch repair were reduced in GFP[hi] DN3 cells (Fig. 3c and Supplementary Data 1). To determine if the gene signature of cell cycle-restricted pre-LSCs was similar to dormant HSCs, we utilized expression data generated from *H2B-GFP* transgenic mice[29] (Supplementary Data 1). GSEA showed that genes downregulated in dormant HSCs were also significantly reduced in cell cycle-restricted pre-LSCs (Fig. 3d and Supplementary Data 1). In contrast, the genes increased in dormant HSCs were not similar to those increased in cell cycle-restricted pre-LSCs (Supplementary Fig. 3c).

**Cell cycle-restricted pre-LSCs acquire *Notch1* mutations.** One of the most striking changes identified in the GSEA was reduced expression in GFP[hi] DN3 cells of genes clustering on different regions of human chromosomes (Supplementary Fig. 4a) that assembled on chromosomes 2 and 15 in mice (Fig. 4a) as compared to GFP[lo] thymocytes. Given that the analysis was performed using the average expression in all samples tested, this striking observation suggested that whole chromosomes were either lost in GFP[hi] or gained in GFP[lo] DN3 thymocytes. Targeted probe analysis for chromosomes 15 and 2 revealed a high frequency of trisomy 15 and 2 in GFP[lo] DN3 thymocytes (Fig. 4b). Genomic PCR for the *IL7r* and *Myc* loci on chromosome 15 (Supplementary Methods) confirmed increased copy number in GFP[lo] cells (Supplementary Fig. 4b). Given the RNA-sequencing (RNA-seq) analysis was performed on pooled samples, we also performed whole-exome sequencing (WES) on GFP[hi] and GFP[lo] DN3 thymocytes isolated from 5 individual doxycycline-pulsed 2-month-old *H2B-GFP;Lmo2*[Tg] mice. In accordance with gene expression data, WES analysis revealed gains of chromosomes 2 and 15 as well as other numerical chromosomal alterations in the GFP[lo] population from all mice analyzed (Fig. 4c). Thus, cycling is associated with aneuploidy in pre-LSCs.

It is postulated that the quiescent state of long-term repopulating HSCs increases the risk of acquired mutations due to the use of error-prone non-homologous end joining-mediated DNA repair[28]. In contrast, cycling HSCs or progenitors can utilize high-fidelity homologous recombination for DNA repair. To define the relationship between cell cycle and mutations in

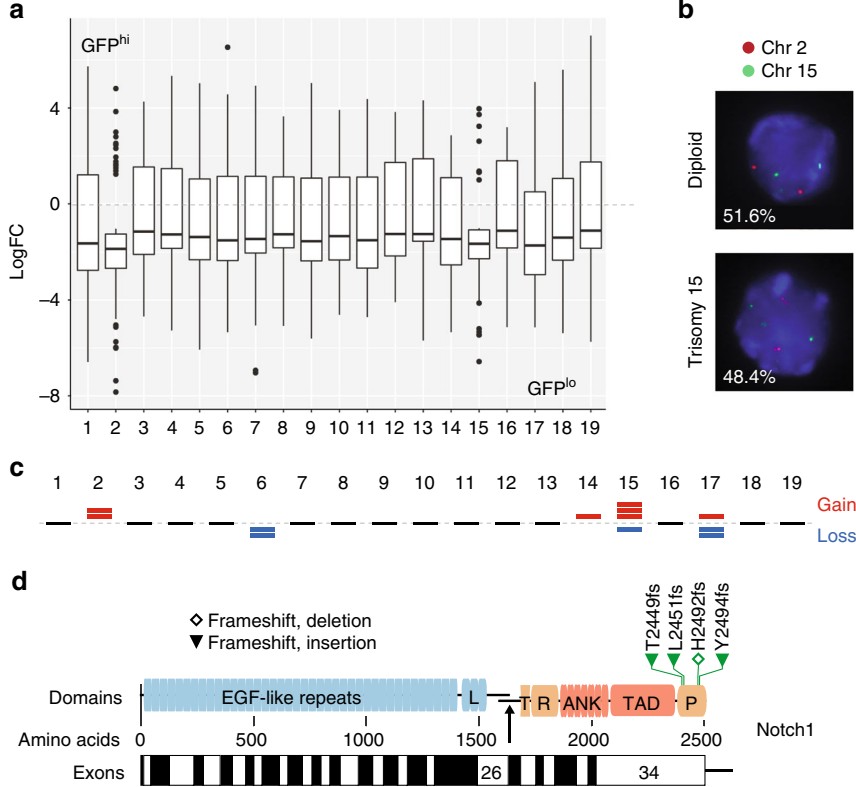

**Fig. 4** Genomic alterations associated with cell cycle in pre-leukemic stem cells (pre-LSCs). **a** Box plot of the relative expression profiling of genes in GFP$^{lo}$ and GFP$^{hi}$ DN3 thymocytes from *H2B-GFP;Lmo2*$^{Tg}$ mice 2 weeks after doxycycline pulse, in accordance with their chromosomal location. Median ± whiskers with maximum 1.5 interquartile range (IQR) and outliers are indicated for each chromosome. Log fold change (LogFC) as compared to median expression among all the samples analyzed. **b** Targeted probe analysis for chromosomes 15 and 2 in GFP$^{lo}$ DN3 thymocytes from *H2B-GFP;Lmo2*$^{Tg}$ mice 2 weeks after doxycycline pulse. Frequency of diploid (top panel) and aneuploid (bottom panel) cells is indicated. **c** Mouse chromosomal ideogram showing the chromosomal gain and losses identified in GFP$^{lo}$ subpopulations of DN3 thymocytes by whole-exome sequencing (WES) on 5 individual doxycycline-pulsed 2-month-old *H2B-GFP;Lmo2*$^{Tg}$ mice. Top red bars represent chromosomal gains and losses are identified by bottom blue bars. **d** Diagram showing the position of mutations found in the *Notch1* gene sequence in purified GFP$^{hi}$ DN3 thymocytes from 2-month old *H2B-GFP;Lmo2*$^{Tg}$ mice (*n* = 5). The amino-acid numbers are shown, with their position in the corresponding exons from the *Notch1* locus. GFP green fluorescent protein, L Lin/ NOTCH repeats, T transmembrane domain, RAM RAM domain, ANK ankyrin repeat domain, TAD transactivation domain, PEST PEST domain; and the arrow indicated the site of cleavage releasing the Notch1 intracellular domain following activation

pre-LSCs, we used the RNA-seq data to identify variants differentially expressed in GFP$^{hi}$ and GFP$^{lo}$ DN3 cells. Overall, we detected 57 genes with variants predicted to be deleterious due to frameshift, splice or premature stop (Supplementary Data 2). While the majority (*n* = 43) were found in both GFP$^{hi}$ and GFP$^{lo}$ cell populations, frameshift mutations upstream of the PEST coding region in *Notch1* were present only in the cell cycle-restricted GFP$^{hi}$ cells (Supplementary Data 2). Targeted sequencing of the *Notch1* locus in GFP$^{hi}$ DN3 thymocytes isolated from 5 individual mice confirmed the presence of activating mutations of *Notch1* in cell cycle-restricted pre-LSCs (Fig. 4d). In addition, a stop in *Tcrg-V3* indicating *Tcrg* gene rearrangement was found exclusively in GFP$^{hi}$ cells. The expression of truncated transcripts of *Tcrg-V3* represents an early marker of clonal selection in cell cycle-restricted pre-LSCs. Thus, distinct mutations occur in pre-LSCs according to their cell-cycle dynamics with recurrent activating mutations of Notch1 limited to cell cycle-restricted pre-LSCs.

**p21 is required for stem cell-like properties of pre-LSCs**. We examined expression of the known inhibitors of cell cycle in HSCs to define the mechanism of cell-cycle restriction in pre-LSCs. Consistent with activation of p53, we found a threefold increase in the expression of *Cdkn1a* (*p21*) in the DN3 GFP$^{hi}$ subpopulation. In contrast, expressions of *Cdkn1b* (*p27*), *Cdkn1c* (*p57*) and the Ink4a family members were not altered in GFP$^{hi}$ cell (Fig. 5a and Supplementary Fig. 5a).

To assess the importance of p21 in cell cycle-restricted pre-LSCs, we generated *H2B-GFP;Lmo2*$^{Tg}$ mice on a *p21*-deficient (*p21*$^{-/-}$) background. Significantly, loss of p21 led to almost complete absence of GFP$^{hi}$ cells by 2 weeks post doxycycline pulse (Fig. 5b and Supplementary Fig. 5b). This loss of cell-cycle restriction was observed in total DN3, where the proportion of cells in G$_0$ was restored to wild-type levels (Supplementary Fig. 5c). Genomic quantification of the *Il7r* and *Myc* loci revealed that *p21* deficiency increased copy number of chromosome 15 in *p21*-deficient *Lmo2*$^{Tg}$ cells lacking p21 (Supplementary Fig. 5d), suggesting that p21 was important for cell-cycle restriction and genomic stability in pre-LSCs. Given the absence of GFP$^{hi}$ cells, we performed serial transplant experiments of total DN3 thymocytes from 2-month-old mice to determine the functional consequences of loss of cell-cycle restriction (Fig. 5b). Interestingly, *Lmo2*$^{Tg}$ DN3 cells lacking p21 were able to generate sevenfold more DN3 progeny than *Lmo2*$^{Tg}$ DN3 cells expressing p21 (Supplementary Fig. 5e). In addition, absence of p21 promoted differentiation to DP cells (Fig. 5d). Despite the enhanced repopulation in primary transplants, serial transplant

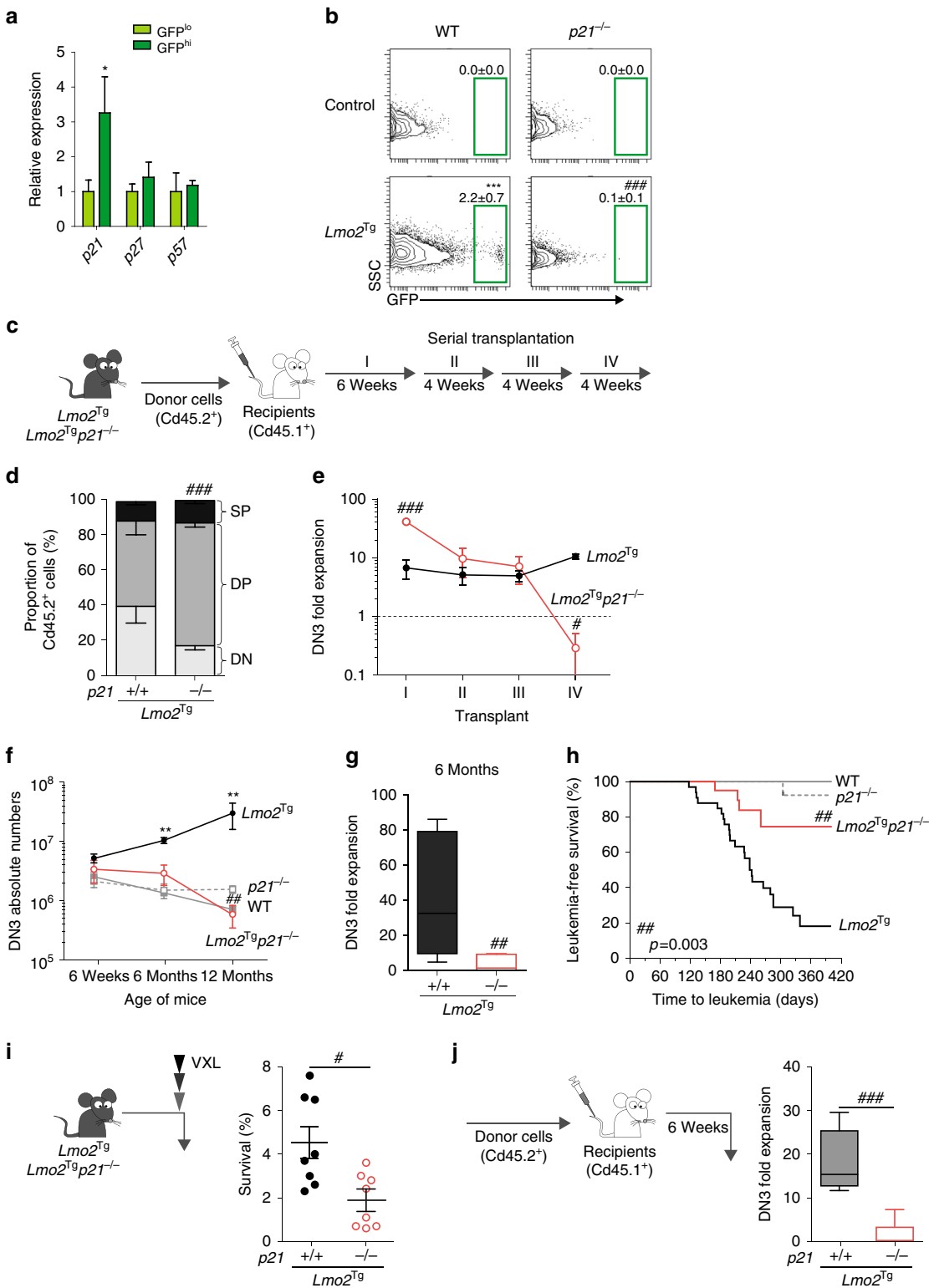

revealed a progressive loss of DN3 repopulating activity such that there was complete loss by the fourth transplant (Fig. 5e). In contrast, the expansion of $Lmo2^{Tg}$ DN3 remained relatively constant over serial transplants.

To determine if this pre-LSC exhaustion was only observable in the setting of proliferative stress related to transplantation, analogous to previous reports with $p21$-deficient HSCs[30], we compared the numbers of DN3 cells in 6- and 12-month-old $Lmo2^{Tg}$ and $Lmo2^{Tg};p21^{-/-}$ mice. As previously reported[31], there

was a progressive increase in the numbers and proportion of DN3 cells in aged $Lmo2^{Tg}$ mice (Fig. 5f and Supplementary Fig. 5f). In contrast, absence of p21 prevented DN3 expansion with aging. Accordingly, there was marked loss of repopulating activity in 6-month-old $Lmo2^{Tg};p21^{-/-}$ thymus compared with age-matched $Lmo2^{Tg}$ mice (Fig. 5g). Consistent with loss of pre-LSCs, there was reduced monoclonality as assessed by $Tcr\beta$ rearrangement (Supplementary Fig. 5g), and most importantly marked reduction of T-ALL penetrance in mice lacking p21 (Fig. 5h). We previously

**Fig. 5** Absence of *p21* promotes loss of cell cycle-restricted, leukemogenic and chemoresistant pre-leukemic stem cells (pre-LSCs). **a** Relative expression of cell-cycle regulators in GFP[lo] and GFP[hi] DN3 thymocytes from *H2B-GFP;Lmo2*[Tg] mice 2 weeks after doxycycline pulse measured by quantitative real-time PCR (qRT-PCR). Gene expression in GFP[lo] cells was used as control, and reported as 1. Values are mean ± s.d., Student's *t*-test *$p < 0.05$. **b** Representative green fluorescent protein (GFP) labeling in DN3 thymocytes from wild-type (WT; $n = 10$), $p21^{-/-}$ ($n = 6$), *Lmo2*[Tg] ($n = 4$) and *Lmo2*[Tg];$p21^{-/-}$ ($n = 5$) mice on a *H2B-GFP* background after 2 weeks of chase. GFP[hi] population framed, mean ± s.d., two-way analysis of variance (ANOVA) test with Tukey's correction. **c** Scheme for serial transplantation into primary (I), secondary (II), tertiary (III) and quaternary (IV) recipients, using pre-leukemic thymocytes from 6-week-old mice. **d** Immunophenotype of donor-derived cells in primary recipients injected with thymocytes from 6-week-old *Lmo2*[Tg] and *Lmo2*[Tg];$p21^{-/-}$ mice. DN, DP and SP represent the CD4−CD8−, double-negative, CD4+CD8+ double-positive and CD4+CD8− single-positive with CD4−CD8+ single-positive populations, respectively. **e** Fold expansion of donor-derived DN3 thymocytes in serially transplanted recipients. **f** Absolute numbers of DN3 T-cell progenitors in the thymus of 6-week (WT, $n = 9$; $p21^{-/-}$, $n = 10$; *Lmo2*[Tg], $n = 9$; *Lmo2*[Tg];$p21^{-/-}$, $n = 10$), 6-month (WT, $n = 5$; $p21^{-/-}$, $n = 5$; *Lmo2*[Tg], $n = 8$; *Lmo2*[Tg];$p21^{-/-}$, $n = 7$) and 12-month-old (WT, $n = 6$; $p21^{-/-}$, $n = 4$; *Lmo2*[Tg], $n = 11$; *Lmo2*[Tg];$p21^{-/-}$, $n = 5$) mice. Two-way ANOVA test with Tukey's correction. **g** Fold expansion of donor-derived DN3 thymocytes from 6-month-old *Lmo2*[Tg] and *Lmo2*[Tg];$p21^{-/-}$ mice in transplanted recipients. Median ± 95% confidence interval (CI), Student's *t*-test, $n = 9$. **h** Kaplan–Meier curves of the time to leukemia for WT ($n = 5$), $p21^{-/-}$ ($n = 24$), *Lmo2*[Tg] ($n = 33$) and *Lmo2*[Tg];$p21^{-/-}$ ($n = 20$) mice. All malignant thymic tumors were diagnosed at necropsy. **i** Proportion of chemoresistant DN3 thymocytes from *Lmo2*[Tg] and *Lmo2*[Tg];$p21^{-/-}$ mice, performed 24 h after the VXL chemotherapy treatment. **j** Fold expansion of donor-derived DN3 thymocytes enumerated in the thymus of recipients injected with thymocytes from *Lmo2*[Tg] and *Lmo2*[Tg];$p21^{-/-}$ mice treated with VXL from (**i**). Values are median ± 95% CI, Student's *t*-test, $n = 12$; **$p < 0.01$, ***$p < 0.001$, as compared to WT; #$p < 0.05$, ##$p < 0.01$, ###$p < 0.001$, as compared to *Lmo2*[Tg] cells, respectively

showed that *Notch1* mutations are acquired during disease progression[20]. To assess the importance of cell cycle in the acquisition of *Notch1* mutations, we used the RNA-seq data to identify variants differentially expressed in *p21*-deficient *Lmo2*[Tg] DN3 cells, and found that *Notch1* mutations were only present in *Lmo2*[Tg] thymocytes (Supplementary Data 2). Targeted sequencing of the *Notch1* locus in DN3 thymocytes isolated from 6-month-old mice revealed that the presence *Notch1* mutations was decreased by twofold in pre-LSCs lacking p21 (Supplementary Fig. 5h). In aggregate, these studies show that p21 was required for clonal evolution and leukemia progression of pre-LSCs.

To assess the role of p21 in therapeutic resistance, we measured repopulating activity in the thymus 24 h after multi-agent chemotherapy. At this time point, the proportion of surviving DN3 thymocytes was fourfold lower in *Lmo2*[Tg] mice lacking p21 (Fig. 5i). Transplant of these chemoresistant thymocytes showed that repopulating activity was maintained in *Lmo2*[Tg] thymocytes but markedly reduced in *p21*-deficient *Lmo2*[Tg] thymocytes (Fig. 5j). The presence of p21 was also important for the resistance of *Lmo2*[Tg] DN3 thymocytes to γ-irradiation (Supplementary Fig. 5i). Thus, p21 mediates resistance of pre-LSCs to chemotherapy and radiation.

Given the lack of cell cycle-restricted DN3 cells in *Lmo2*[Tg];$p21^{-/-}$ mice, we isolated total DN3 cells for gene expression profiling to gain insight into the role of p21 in pre-LSCs. We restricted studies to 6-week-old wild-type, $p21^{-/-}$, *Lmo2*[Tg] and *Lmo2*[Tg];$p21^{-/-}$ mice, an age prior to loss of repopulating activity in *p21*-deficient *Lmo2*[Tg] mice (Fig. 5g). Principal component analysis confirmed clustering according to genotype (Fig. 6a). Comparison of wild-type DN3 cells with $p21^{-/-}$ DN3 thymocytes revealed minimal changes with only 3 genes differentially expressed more than twofold (Supplementary Data 3). To determine how p21 abrogates *Lmo2*-induced leukemogenesis, we compared *Lmo2*[Tg] DN3 cells with *Lmo2*[Tg];$p21^{-/-}$ DN3 thymocytes. Importantly, there was no difference in the expression of Lmo2 or its downstream targets responsible for self-renewal[19,32,33] such as Lyl1, Hhex and c-Kit (Fig. 6b). Overall, there were 463 differentially expressed genes: 153 increased and 310 decreased more than twofold in *Lmo2*[Tg];$p21^{-/-}$ DN3 cells (Fig. 6c and Supplementary Data 3). Gene ontology pathway analysis showed that the reduced genes were enriched for general metabolic pathways of transcription and translation as well as signaling (nuclear factor-κB, mitogen-activated protein kinase), G1/S transition and apoptosis (Supplementary Data 3). These changes were confirmed using GSEA, which revealed a striking reduction in genes involved in DNA replication, splicing and the

proteasome (Fig. 6d). Importantly, these changes were not seen with *p21*-deficient DN3 cells compared with wild-type DN3 thymocytes (Supplementary Fig. 6a). Thus, *p21*-mediated cell-cycle restriction was required for widespread metabolic processes in the context of oncogene-transformed cells.

To understand the cellular fate of *Lmo2*[Tg] DN3 cells in the absence of p21 (apoptosis or differentiation), we co-cultured sorted DN3 thymocytes on OP9-DL1 stroma cells, which support in vitro division and differentiation of T-cell progenitors[34]. Using this approach, we confirmed that absence of p21 promoted the differentiation of *Lmo2*[Tg] DN3 thymocytes into DP cells (Fig. 6e). Pre-LSCs develop just prior to the β-selection checkpoint during which T-cell fate is tightly regulated by asymmetric cell division (ACD)[35], a homeostatic cell division process also crucial for self-renewal of HSCs[36,37]. ACD can be observed by the polarized segregation of the "differentiation fate determinant" Numb in dividing cells, which generate one identical immature/stem and one differentiated daughter cell[35,38,39]. Given that p21 has been associated with a switch from asymmetric to symmetric division in co-cultured stem cells[40], we examined the frequency of ACD in sorted DN3 thymocytes using the Numb distribution in dividing cells (Fig. 6f). Consistent with the stem cell-like phenotype of pre-LSCs, the frequency of ACD was significantly increased in *Lmo2*[Tg] DN3 cells compared with wild-type and $p21^{-/-}$ DN3 cells (Fig. 6f and Supplementary Fig. 6b). The increased ACD observed in *Lmo2*-expressing DN3 cells was significantly reduced in the absence of p21, restoring the preponderance of symmetric division observed in wild-type thymocytes. Altogether, these results show that *p21*-deficiency promotes differentiation of pre-LSCs at the expense of ACD, which correlates with the importance of p21 for the maintenance of self-renewing pre-LSCs during leukemia development.

## Discussion

Pre-LSCs are an important cell population during leukemic development, and provide a pool of cells that give rise to relapse[2,3,41]. The clinical relevance of pre-LSCs has recently been highlighted by sensitive molecular assays, where patients with detectable pre-LSCs during complete remission have an increased risk of relapse[5,42]. However, the properties of pre-LSCs that allow them to escape high-dose chemotherapy remain unknown. In this paper, we show that clonal evolution and therapeutic resistance can be defined by their cell-cycle characteristics. Specifically, we identify the presence of a rare subpopulation of cell cycle-restricted pre-LSCs that have enhanced therapeutic resistance and most importantly represent the population of cells that acquire

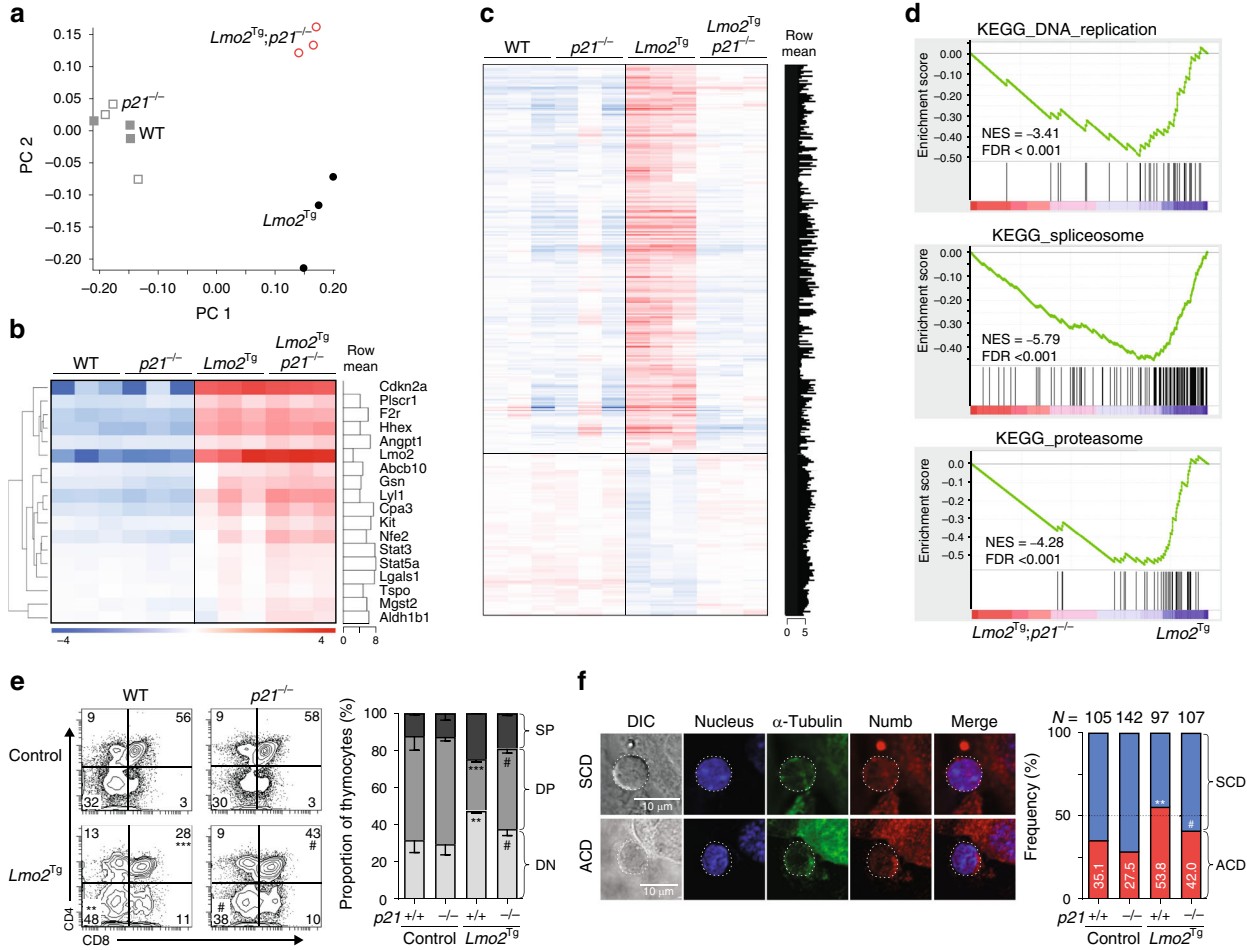

**Fig. 6** *p21* is crucial for the stem cell-like program, p53-mediated response and block in differentiation. **a** Principle component analysis (PCA) of the gene expression signatures in purified DN3 thymocytes from 6-week-old wild-type (WT), *p21*−/−, *Lmo2*Tg and *Lmo2*Tg;*p21*−/− mice. **b** Heat map of *Lmo2*-associated upregulated gene signature in purified DN3 thymocytes from (**a**). **c** Heat map of genes differentially expressed (FDR < 0.05) in DN3 thymocytes from 6-week-old *Lmo2*Tg, as compared to *Lmo2*Tg;*p21*−/− as well as wild-type (WT) and *p21*−/− controls. Row mean: relative expression of each gene as compared to the average expression for each genotype analyzed. **d** Gene set enrichment analysis (GSEA) plot of DNA replication, ribosome function (spliceosome) and proteasome genes in DN3 thymocytes from *p21*-deficient *Lmo2*Tg mice, as compared to *Lmo2*Tg mice (top panels), as well as *p21*−/− DN3 cells compared to wild-type (WT) DN3 thymocytes (bottom panels). FDR false discovery rate, NES normalized enrichment score. **e** Immunophenotype of co-cultured DN3 thymocytes for 5 days for in vitro differentiation assays. $N = 2$ independent experiments, mean ± s.e.m., Student's *t*-test. **f** Representative confocal immunofluorescence of undergoing symmetric cell division (SCD, top row) and asymmetric cell division (ACD, bottom row) of DN3 thymocytes from 6-week-old mice in co-culture assays. Sorted DN3-stromal cell conjugates were fixed and co-stained with α-Tubulin (red) and the cell-fate protein Numb (green). $N = 2$ independent experiments, with specific numbers of cells analyzed indicated. Chi-Square test described for each genotype was performed (Supplementary Fig. 6b); **$p < 0.01$, ***$p < 0.001$, as compared to WT; #$p < 0.05$, as compared to *Lmo2*Tg cells, respectively

oncogenic *Notch1* mutations necessary for clonal evolution to T-ALL. By genetic deletion of the cell-cycle inhibitor p21, we show that overcoming cell-cycle restriction abrogates this therapeutic resistance and significantly reduces clonal evolution of pre-LSCs. Thus, we show that cell-cycle restriction is a fundamental property of pre-LSCs that explains their long-term competitive advantage and potential for causing relapse following high-dose chemotherapy.

We have previously shown that Lmo2 induces aberrant self-renewal of immature T-cell progenitors without preventing T-cell differentiation[19], and as such display features typical of pre-LSCs[43]. We now extend these findings to show that long-term self-renewal necessary for clonal evolution is limited to a rare subpopulation of cell cycle-restricted pre-LSCs. The impaired repopulating activity of GFP^lo DN3 thymocytes might be explained by increased cycling leading to impaired homing. However, this is highly unlikely as DN3 cells lacking p21 had increased repopulating activity in primary transplants

(Fig. 5e) despite increased cycling. Consistent with their stem cell-like properties, these cells are also more resistant to irradiation and combination chemotherapy. We did not directly examine the leukemic potential of the GFP^hi cells enriched by chemotherapy; however, Ebinger et al.[11] recently showed that label-retaining cells surviving chemotherapy in human B-ALL xenografts retained leukemia-initiating potential. In sharp contrast with normal HSCs following stress-induced proliferation[25,44,45], we show that once pre-LSCs enter a proliferative state, they are unable to return to a cell cycle-restricted state (Fig. 2e). Thus, strategies that promote cell cycle prior to chemotherapy may be able to eradicate pre-LSCs without detrimental effects on normal HSCs.

The role of p21 in leukemogenesis is controversial with both tumor suppressive and promoting activity. For example, knockdown of *p21* in MLL-AF10-induced leukemia accelerated disease[46]. In contrast, PML/RAR (promyelocytic leukemia/retinoic acid receptor)-transformed HSCs required p21 for long-term

**Table 1 Primers used for genomic and quantitative PCR**

| Gene | Name of the primer | Sequence |
|---|---|---|
| Cdkn1a (p21) | mCdkn1a_Fw | tacccgtctttccccagagt |
|  | mCdkn1a_Rv | acactgtcacccacagatcg |
| Cdkn1b (p27) | mCdkn1b_F | gttagcggagcagtgtcca |
|  | mCdkn1b_R | tctgttctgttggcccttt |
| Cdkn1c (p57) | mCdkn1c_F | caggacgagaatcaagagca |
|  | mCdkn1c_R | gcttggcgaagaagtcgt |
| **TcrB rearrangement** | **Name of the primer** | **Sequence** |
| mTcrb | S16 Fw | AGGAGCGATTTGCTGGTGTGG |
|  | S16 Rv | GCTACCAGGGCCTTTGAGATG |
|  | Tcr Vb5 Fw | CCCAGCAGATTCTCAGTCCAACAG |
|  | Tcr Vb8 Fw | GCATGGGCTGAGGCTGATCCATTA |
|  | Tcr Jb2 Rv | TGAGAGCTGTCTCCTACTATCGATT |
| **Genomic PCR of loci on Chr. 15** | **Name of the primer** | **Sequence** |
| mll7r | mll7r Fw | ACTGTGGTTGGGTGCCTTAG |
|  | mll7r Rv | AAATTGGCAGAACCAACCAG |
| mMyc | mMyc Fw | ACACGGAGGAAAACGACAAG |
|  | mMyc Fw | TCGTCTGCTTGAATGGACAG |
| **Notch1 sequencing** | **Name of the primer** | **Sequence** |
| mNotch1 | Exon26_4853Fw | CCCTGTATGACCAGTA |
|  | Exon26_5250Rv | ACGGATGTCCATGGGGT |
|  | Exon27_5253Fw | CTCCATTGTCTACCTG |
|  | Exon27_5401Rv | TCTTCACGGCCTCAATC |
|  | Exon34_6423Fw | TCCCCTGTTCCTGGCCG |
|  | Exon 34_6548Fw | TGCACCACGATATCGTG |
|  | Exon34_7265Fw | ATAGCATGATGGGGCCACTA |
|  | Exon 34_6849Fw | ACCCCATGGCTACTTGTCAG |
|  | Exon34_8157Rv | CTT CAC CCT GAC CAG GAA AA |
|  | Exon34_7284Rv | TAGTGGCCCCATCATGCTAT |
|  | Exon34_7675Rv | CCACAGGGGAGGAGGAGTAA |
|  | Exon34_8194Rv | TGA ATC CTT GTT CAT ATT TTA CAG ACA CAC AGG G |

self-renewal[47]. The observation that Lmo2[Tg] DN3 cells lacking p21 have enhanced primary repopulating capacity but loss in subsequent transplants (Fig. 5f) provides one possible explanation for this controversy. The properties identified for cell cycle-restricted pre-LSCs have many parallels with normal HSCs under conditions of stress or aging. First, p21 is only important for HSCs in the setting of stress[30]. Second, activation of the p53–p21 axis promotes cell-cycle arrest and DNA repair in irradiated HSCs[28,48]. Third, error-prone DNA repair occurs in stress or aging HSCs due to reduced expression of genes required for high-fidelity homologous recombination and components of the MCM helicase. Thus, we propose that cell cycle-restricted pre-LSCs arising from a committed progenitor behave like normal HSCs following DNA damage.

Mutation analysis of pre-LSCs identified an intriguing relationship between cell cycle and types of genomic mutations. Cell cycle of pre-LSCs was associated with a high frequency of aneuploidy. Almost half of all cells undergoing cell division had trisomy 15 and/or 2 (Fig. 4a). Although it is difficult to know which comes first (aneuploidy or cell cycle), the higher rate of aneuploidy in pre-LSCs unable to arrest (Lmo2[Tg];p21[−/−]) suggests that cell cycle induces aneuploidy. Furthermore, studies of cell lines with trisomy generated from transgenic mice suggest that aneuploidy slows rather than promotes cell cycle[49,50]. Trisomy 15 has been reported in two other mouse models of leukemia[51,52], suggesting the selective advantage for numerical abnormalities of chromosome 15 occurs irrespective of the oncogene or cell lineage. Acquisition of an extra copy of c-Myc is one possible explanation[51]. Decreased expression of MCM helicases, together with their reported interaction with LMO2[53], may explain the high frequency of aneuploidy. The other striking difference was the presence of Notch1 mutations in GFP[hi] cells. Given these cells have a long-term selective advantage, and Notch1 enhances self-

renewal[22], this observation provides an explanation for the high frequency of Notch1 mutations in T-ALL.

Previous studies have shown that environmental cues from the niche control the balance between symmetric and asymmetric divisions of HSCs and this balance can be perturbed by the expression of oncogenes[39,54]. Using the segregation of Numb—a negative modulator of Notch1 signaling—to assess division patterning in thymocytes, we found that expression of the Lmo2 oncogene significantly increased the frequency of asymmetric division in DN3 cells. Conversely, absence of p21 reduced asymmetry and promoted differentiation in dividing Lmo2[Tg] DN3 cells, restoring a division patterning similar to wild-type thymocytes. Thus, cell cycle plays a crucial role in the decision between different types of division, which ultimately effects the maintenance of the pre-LSC population. Given absence of p21 also increases aneuploidy, one unifying possibility is that the shortened duration of cell cycle leads to missegregation of differentiation fate determinants and chromosomes in dividing cells, ultimately impairing the stem cell-like properties of pre-LSCs.

In conclusion, our work provides the first in vivo evidence that cell-cycle restriction is essential for self-renewal, clonal evolution and therapeutic resistance of pre-LSCs in a murine model of T-ALL. We also demonstrate that pre-LSCs fundamentally diverge from normal HSCs with regard to their ability to return to a cell cycle-restricted state following stress-induced proliferation. We propose the H2B-GFP system will be a powerful in vivo model to determine if similar properties apply to other models of pre-LSCs and identify and test strategies to overcome quiescence of relapse-inducing cells.

## Methods

**Mouse experiments**. All experiments were approved and complied with the ethical regulations mandated by the AMREP Animal Ethics Committee. The

current study was performed using the previously described *TetOP-H2B-GFP*[KI/+], kindly given by H. Hock from the Harvard Stem Cell Institute[14], *p21*-deficient (*p21*[−/−]) and the *CD2-Lmo2* (*Lmo2*[Tg])[31] mouse models. All mouse lines were backcrossed onto a C57BL/6J background for 10 generations and maintained in pathogen-free conditions according to institutional animal care guidelines.

**FACS analysis.** Flow cytometry analysis and cell sorting were done as previously described[19,20] on single-cell suspensions of T-cell progenitors. Thymocytes were stained using BD Pharmingen antibodies (BD Australia, North Ryde, NSW) against mouse CD4 (1:500; RM4-5), CD8 (1:500; 53-6.7), CD25 (1:400; PC61.5), CD44 (1:500; IM7), CD45.1 (1:125; A20), CD45.2 (1:125; 104), Thy1.2 (1:500; 53-2.1) and TCRβ (1:500; H57-597) to describe T-cell populations in steady state or after transplantation. Cell-cycle analysis was performed as described previously[20], using an antibody against Ki67 (1:10; BD Australia, Cat. no. 556027) or the isotype control, and staining DNA using DAPI (Sigma-Aldrich). Apoptosis was measured using the BD Pharmingen antibody against AnnexinV (1:40; Cat. no. 556420) and the permeable nucleic acid dye 7-aminoactinomycin D (BD Australia) following the manufacturer's protocol. FACS analysis was performed using a LSRII and a LSR Fortessa cytometers and cell sorting was performed with a FACSAria or BD Influx (BD Australia, North Ryde, NSW).

**Modeling cell-cycle kinetics.** The modeling cell-cycle kinetics was performed using the absolute numbers of GFP[hi] and total DN3 cells from 6-week-old H2B-GFP;*Lmo2*[Tg] mice and littermate controls after 6 weeks of Doxycycline pulse, followed by 0, 1, 2, 4 or 8 weeks of chase, as indicated in Supplementary Fig. 1A. Ratios were formed as GFP[hi]_DN3/DN3 and in order to stabilize the variance, the log (base 10) of the ratios was calculated as follows: logratio = $\log_{10}$ ((GFP[hi]_DN3 + 100)/DN3). A constant (=100) was added to the numerator in the ratio to avoid taking the log of zero. The constant was chosen to be less than the smallest reported non-zero value of GFP[hi]_DN3 (=364). The combined data from both groups were fitted in an exponential model as $Y = A + B*(R**X)$ in which in which $Y$ is the $\log_{10}$ of the ratio at week $X$. The nonlinear parameter ($R$) was constrained to be <1. The model has the property that at time zero (i.e., $X = 0$), $Y = A + B$ and as time increases, $Y$ asymptotes at $A$. The FITCURVE procedure in the GenStat statistical package[55] was used to fit increasingly complex models (to account for different values of the parameters in each group (H2B-GFP;Lmo2 and H2B-GFP). The model fitting exercise indicated that separate intercept "$A$" parameters were required for each group ($p < 0.001$) and separate slope coefficient "$B$" parameters were also required for each group ($p < 0.001$). The nonlinear parameter "$R$" was not significantly different between the groups ($p = 0.124$); nevertheless, separate nonlinear parameters were retained in the final model. The adjusted $R^2$ value for the final model was 83.8%. The fitted model is shown graphically in Fig. 1c. Bootstrapping the residuals from the fitted models was used to test for significant differences between the groups in their times for achieving 1-log and 2-log reductions in the ratios ($n = 5000$ bootstrap samples were used).

**Transplantation assays.** Transplantation assays were performed by intravenously injecting thymus cells into sublethally irradiated (650 Rads) isogenic Ly5.1 (Cd45.1) mice. Leukemic mice were scored positive when they presented signs of overt leukemia, which was confirmed at necropsy. Kaplan–Meier survival and statistical analysis were performed using GraphPad Prism 6.0 software (GraphPad Software Inc., San Diego, CA).

**RNA analysis.** Total RNAs for global gene expression were prepared from a pool of 0.7–1.0 × 10[6] sorted cells from 3 to 5 different 6-week-old mice using RNeasy extraction kit (Qiagen, Chadstone, VIC, Australia). Global RNA amplification and hybridization to the murine Agilent gene expression array was performed by the Australian Genome Research Facility Ltd (AGRF, Parkville, VIC, Australia) and analyzed using Agilent In Situ Microarray algorithm. Differentially expressed genes between DN3 thymocyte samples were determined using a linear model and the empirical Bayes method previously described[56]. Data for RNA-seq are available at: https://www.ncbi.nlm.nih.gov/geo/query/acc.cgi?acc=GSE110132.

For gene expression experiments, total RNAs were prepared from 50,000 to 100,000 sorted cells from 6-week-old mice using RNeasy extraction kit (Qiagen, Chadstone, VIC, Australia). First-strand complementary DNA (cDNA) synthesis was performed by reverse transcription as previously described[57]. Primer sequences used are listed in Table 1. Real-time quantitative PCR was done with SYBR Green Master Mix (Applied Biosystems—Life Technologies Australia, Mulgrave, VIC) on Roche LightCycler480 II (Roche Diagnostics Australia, Castle Hill, NSW). Delta delta Ct values were calculated by using Ct values from *Hprt* and *β-actin* genes as reference.

**Whole-exome sequencing.** Genomic DNA samples were extracted from sorted GFP[hi] and GFP[lo] DN3 cells from 5 individual 2-month-old *H2B-GFP*;*Lmo2*[Tg] mice using the QIAGEN DNeasy Blood & Tissue Kit (Qiagen, Chadstone, VIC, Australia). DNA quantity and quality were measured using a NanoDrop ND-1000 spectrophotometer (Thermo Fisher Scientific Australia Pty Ltd, Scoresby, VIC, Australia). Genomic DNA amplification was performed on high-quality genomic DNA (300–1500 ng) using DETAILS by Macrogen Inc. (Macrogen Oceania, Sydney,

Sydney, NSW, Australia). Briefly, library construction was prepared by random fragmentation of the DNA or cDNA sample, followed by 5' and 3' adapter ligation. Library construction used DNA tagmentation, which combined fragmentation and ligation reactions into a single step, for increasing the efficiency of the library preparation process. Adapter-ligated fragments were then amplified by PCR and purified on gel. The whole exome was captured through target enrichment of DNA samples and construction of a hybridization library, using the Agilent SureSelectXT Library Prep Kit (Agilent Technologies, Santa Clara, CA, USA) according to the manufacturer's instructions. Sequencing was done with HiSeq 4000 instruments in high-output mode with TruSeq 3000 4000 SBS v3 chemistry. All runs were 101-nt paired-end reads, and data were analyzed with the HSC v3.3 software. Raw data were generated by the Illumina HiSeq 4000, which utilized HiSeq Control Software v3.3 for system control and base calling through the Real Time Analysis v2.7.3 software. The base calls binary was converted into FASTQ utilizing the Illumina bcl2fastq v2.17.1.14 protocol. Exome sequencing was performed on amplified DNA samples from sorted GFP[hi] and GFP[lo] DN3 cells from individual 2-month-old *H2B-GFP*;*Lmo2*[Tg] mice by Macrogen Inc. (Macrogen Oceania, Sydney, NSW, Australia).

**Chemoresistance assays.** For chemoresistance assays, 6-week-old mice were intraperitoneally injected with induction therapeutic regimen (VXL) consisting of 0.15 mg/kg vincristine (Vincristine Sulfate, Pfizer), 5 mg/kg dexamethasone (DBL Dexamethasone, Hospira) and 1000 U/kg L-asparaginase (Leunase, Sanofi-Aventis)[27]. Thymocytes were harvested, counted and characterized by flow cytometry 24 h after treatment with the VXL regimen.

**In vitro differentiation assays.** For co-culture assays, 100,000 sorted DN3 cells were washed, and seeded onto OP9-DL1 stromal cells, as previously described[22]. Briefly, thymocytes were co-cultured on stromal cells in reconstituted alpha-minimum essential medium (12561, Gibco, Life Technologies, Scoresby, VIC, Australia) supplemented with 10% heat-inactivated fetal bovine serum (12318, Gibco), 10 mM HEPES (15630-060, Gibco), 1 mM sodium pyruvate (11360-070, Gibco), 55 μM β-mercaptoethanol (21985-023, Gibco), 2 mM Glutamax (15750-060, Gibco), penicillin/streptomycin (15140-122, Gibco), 5 ng/mL FLT-3 Ligand (308-FK-025, R&D Systems, Minneapolis, MN, USA) and 5 ng/mL IL-7 (217-17, PeproTech, Rock Hill, NJ, USA). For differentiation assays, cells were harvested, counted and characterized by flow cytometry after 5 days of co-culture.

**Cell division patterning assays.** For cell division patterning assays, 7 × 10[3] OP9-DL1 stromal cells were first plated onto glass-bottom 8-well culture chamber slides (Thermo Fisher Scientific, Life Technologies, Scoresby, VIC, Australia) and left to adhere overnight, as previously described[35]. Then, 1–4 × 10[4] sorted DN3 thymocytes were added with fresh media and co-cultured for 24–48 h. DN3-stromal cell conjugates were washed and fixed as previously described[58]. Cells were blocked using phosphate-buffered saline (PBS) 1 × + 1% w/v bovine serum albumin (BSA), for 20 min at room temperature, and incubated for 30 min at room temperature in PBS 1 × + 0.25% v/v Triton X-100 for permeabilization. Cells were stained in PBS 1× + 2% w/v BSA with anti-Numb (1:100; ab4147, Abcam, Sapphire Bioscience Pty. Ltd., Redfern, NSW, Australia) and anti-Tubulin (1:100; sc-32293, Santa Cruz Biotechnology, Dallas, TX, USA). Cells were washed twice using Perm/Wash buffer (BD Australia, North Ryde, NSW) and incubated in permeabilization buffer with donkey Alexa Fluor 488-conjugated anti-mouse (1:300; A-21202, Molecular Probes, Life Science) and donkey Alexa Fluor 546-conjugated anti-goat (1:300; A-11056, Molecular Probes, Life Science) secondary antibodies for 1 h on ice. Cells were washed, incubated with 1 ng/mL DAPI for 15 min, washed twice and mounted using Mowiol mounting medium (81381, Sigma-Aldrich Pty Ltd, Castle Hill, NSW). At least 20 different fields of view were collected for blinded quantification, for each time point. All images were acquired using a Nikon A1r Plus SI inverted confocal microscope (Nikon Australia, Rhodes, NSW) using a Plan Apo 60× oil objective.

## Data availability

The datasets generated during the current study are available in the Gene Expression Omnibus repository at: https://www.ncbi.nlm.nih.gov/geo/query/acc.cgi?acc=GSE110132. All relevant data that support the findings of this study are available from the corresponding author upon reasonable request.

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

## Acknowledgements

We thank Geza Paukovics, Jeanne LeMasurier, Eva Orlowski-Oliver and Phil Donaldson from the AMREP Flow Cytometry Facility for assistance with flow cytometry. We also thank Stephen Cody and Iśka Carmichael from the Monash Micro Imaging platform as well as Shilpa Bereeka and Loretta Cerruti for technical assistance. We thank Associate Professor Steven Lane from the QIMR Berghofer, Professor Sarah Russell and Dr Mirren Charnley from the Peter MacCallum Cancer Centre and the Centre for Micro-Photonics of the Swinburne University of Technology, as well as Dr Ian Majewski and Dr Kim Pham from the Walter and Eliza Hall Institute for Medical Research (WEHI) for assistance with experimental design and intellectual input. This work was supported by Project Grants (1052313 (to D.J.C.), 1047554 (to S.B.T.)) and a Career Development

Fellowship (1047630 (to S.B.T.)) from the Australian National Health and Medical Research Council (NHMRC), and a Senior Medical Research Fellowship from the Sylvia and Charles Viertel Foundation (to D.J.C.).

## Author contributions

C.S.T. and D.J.C. designed research, analyzed data and wrote the manuscript; C.S.T., J.S., S.K.C., S.G., A.N.G., A.A.G., S.E.S. and N.L. performed research and analyzed data; P.K., S.B.T. and D.R.P. designed research and analyzed data; N.C.W., K.T., F.Y. and J.R. analyzed data; and S.M.J. provided reagents, designed research and reviewed the manuscript.

## Additional information

**Competing interests:** The authors declare no competing interests.

