## [Peer Review File · Nature Communications]

Reviewers' comments:

Reviewer #1 (Remarks to the Author):

In this paper the authors show that restricted cell cycle in pre-LSCs is essential for clonal evolution and therapy resistance. Using an elegant mouse model they follow pre-LSC in a LMO2 tg mice that were backcrossed with an inducible HB@-GFP and show that DN3 that do not proliferate have a increase self-renewal capacity able to serially transplant compared th cuycing DN3-GFP^{low} cells. They also show that these low cycling DN3 pre-LSC are more resistant to chemotherapy and are responsible for clonal evolution.

There is nevertheless some concerns that will need to be addressed:

-Fig 2: The fact that non-cycling cells are engrafted better has been well-documented in HSC. For thge DN3, it will be important that the decrease in engraftmnet is not due to lack or decrease in homing.

Could the authors explained why they used a 4 week setting for their serial transplant.

- Fig 2e: It is unclear why the DN3-GFP^{low} cells are not reacquiring GFP. They should divided as they show at leats in first transplant their capacity to engraft. Is that means that a 6 weeks chase is too long and that most of these cells have lose GFP. Soem early chasing ptime should be provide to show that it is not an artefact. Also looking at the cell cycle status of the GFP^{low} and GFP^{high} after tatsnplant will be helpful.

- In Fig 2S it will be of interest to show the proportion of DN3 cells athta are GFP⁺ versus GFP⁻

- After chemotherapy, an increaese of DN3 GFP⁺ fraction compared to DN3 GFP⁻ but only from 9 to 2 %. It will be more informative to show the frequency of DN3GFP⁺ and GFP⁻ before and after chemotherapy. Are GFP⁺ really enriched after chemotherapy and is this increaese significant.

- They the show that GFP+DN3 have an increaese in p21 and by using p21KO demonstrate that DN3 pre-LSC loose their engraftment potential. I suppose the same will happened if inducing the GFP⁺ cells into cycle by cytokine stimulation like possibly in this case IL.2. This will be more clinically relevant.

All in all it is an interesting paper that provide dierct evidence of teh potential role of dormant pre-LSCs.

Reviewer #2 (Remarks to the Author):

Tremblay and colleagues analyze a novel mouse line that allows non-dividing cells to be tracked (based on H2B-GFP label retention) in a T-ALL model (Lmo2 transgenic). In this mouse model, they classify a subset of DN3 cells that are non-dividing to be pre-leukemic stem cells, which show serial transplantation capacity as well as greater chemotherapy resistance. Cell cycle inhibitor p21 was more highly expressed in this cell population, and by crossing with p21-KO mice, the authors demonstrate that loss of p21 induces cell division and differentiation, limiting leukemic capacity. While this is an interesting and novel study, several aspects of the work need to be address before it should be published, as outlined below.

Major comments:

1. How do the authors explain the 35% G0 cells in the GFP^{lo} population in Figure 1E? Are these also cell-cycle restricted pre-LSCs?
2. Figure 2 demonstrates that GFP^{hi} DN3 cells can be serially transplanted while GFP^{lo} cells cannot. However, it is unclear whether GFP^{hi} cells can still be observed in these recipients. If the serially-transplantable leukemia come from proliferative cells (even if originally derived from quiescent cells), it questions the conclusion of the manuscript.
3. Figure 2F shows that 9% of GFP^{hi} DN3 cells can survive chemotherapy. To demonstrate this is a useful model of pre-LSCs/disease relapse, it would be important to show that these cells are not

senescent and are still functional pre-LSCs (or LSCs) and can drive disease progression in host mice or recipients (and that post-treatment leukemia does not arise from the GFPlo cells).

4. Figure 4C/D nicely summarizes the gain and loss of chromosomes in GFPlo cells and Notch mutations in GFPhi cells. Are these mutations also found following serial transplantation of GFPhi cells (or in the final T-ALL)? This could help to understand disease progression.

5. Related to the above, the authors describe Lmo2-expression as the “pre-leukemic” mutation and that accumulation of additional mutations leads to leukemia. However, they then describe additional genetic abnormalities in pre-LSCs. At what point do these cells become LSCs or leukemia?

6. Given the availability of small molecule p21 inhibitors and claimed therapeutic relevance of the findings, it would be important to know whether p21 inhibitors can replace genetic deletion of p21 to induce differentiation and/or prevent T-ALL.

7. Leukemias (and associated LSCs) are heterogeneous group of cancers. The authors should therefore be very careful about drawing broad conclusions from their results derived from one T-ALL mouse model. For example, with the current data, it is inappropriate to claim that “we show that cell cycle restriction is a fundamental property of pre-LSCs”. Similarly, the title should accurately reflect the results.

8. Revisions to the Figures are necessary to (1) highlight the key information (e.g. is Figure 3 necessary?) and (2) easy to follow (e.g. this reviewer found the layout of Figure 5 confusing).

Minor comments:

9. Throughout: The comparisons of DN3 pre-LSCs to HSCs throughout the manuscript is confusing and should be removed (or more carefully described).

10. Intro: This reviewer is unsure whether the LSC concept is limited to “rare cell population”.

11. Intro: Given the background to pre-LSCs in leukemias, it would be appropriate to also discuss recent observations of clonal hematopoiesis.

12. Intro: It would also be appropriate to introduce the literature on Lmo2 in T-ALL.

13. Figure 1B: Based on the negative population, flow cytometry voltages differ between experiments but the positive gate remains the same.

14. Figure S2D clearly shows an increase in numbers of GFPhi DN3 cells in older mice and those with T-ALL. But it is not clear when the pulses of Dox were provided. Where do these cells come from? Are they generated by differentiation or through self-renewal of DN3 cells? Additionally, given the age-related changes in lymphoid production, it would be important to compare this with the WT setting.

15. Figure 4B and/or legend needs to be corrected.

16. Figure S6 legends need to be corrected.

17. Figure 6D – quantification of the biological repeats of this assay should be plotted. It appears p21KO does not completely rescue the Lmo2 phenotype in vitro.

18. Figure 6E – images displayed are difficult to understand. These should be either enlarged/higher resolution or moved to a supplementary figure and the data summary displayed alone

19. It may be more appropriate to let others describe this work as “elegant”, rather than be one of the authors own conclusions

Reviewer #3 (Remarks to the Author):

Tremblay et al examine cell cycle kinetics of pre-LSC by using their previously published Lmo2 transgenic T-ALL mouse model crossed with a doxycycline inducible H2B-GFP mouse and later with p21^{-/-} mice.

Their model allows examination of GFP expression in dividing cells. Using the previously determined pre-LSC population of CD4⁻CD8⁻CD25⁺CD44⁻ T cells (DN3 cells) the authors show that in Lmo2Tg mice the population of GFPhi cells remains significantly higher than in control mice

and that this population resides to a large part in the G0 phase of the cell cycle, indicating that these cells are cell-cycle restricted. By serially transplanting Lmo2Tg pre-LSC into irradiated CD45.1+ recipient mice, the authors can nicely show that the repopulating capacity is contained only in the expanding GFP^{hi} cell population which is due in part to increased differentiation of T cells in GFP^{lo} population. In addition, secondary, tertiary and quaternary recipients of GFP^{hi} DN3 cells developed T-ALL, indicating acquisition of additional genetic events in pre-LSC. The authors then combine gene expression profiling and functional assays to identify p21 as a crucial pre-LSC regulating gene.

The question of pre-LSC functional properties and how to target these cells for durable elimination of leukemia is an important and enduring question in the field. As such, the establishment of this model and the performed experiments significantly contribute to understanding how pre-LSC may differ from normal HSC. Gene expression analyses and WE sequencing additionally provide a valuable data platform for further studies.

The experiments are sophisticated and well executed and sufficiently described. The authors have performed an extensive amount of experiments which are very nicely presented in the Figures. They attempt to tie together pre-LSC properties, p21, aneuploidy and Notch in their discussion of their data. To me, these are separate observations and while very valid however are not entirely convincingly shown to be completely dependent on or a consequence of one another.

Minor Queries:

1. FIGURE 6b and supplementary Fig6a: differential gene expression between Lmo2Tg and Lmo2Tgp21^{-/-} are much better visualized i.e. more convincing in the supplementary figure, would perhaps suggest to include in main manuscript.
2. p21 is crucial for function of pre-LSC and pre-LSC with GFP^{hi} properties acquire Notch mutations. The authors postulate that this is due to DNA repair mechanisms. Are Notch mutations also found in the Lmo2Tgp21^{-/-} mice? Did the authors specifically look for this? Is acquisition of Notch mutations only a function of cell cycle?
3. Chromosome additions are acquired in GFP^{lo} populations and thus postulated to be dependent on cell cycle. The authors speculate on mechanisms but provide no evidence. Again, do the Lmo2Tgp21^{-/-} mice exhibit aneuploidy?
3. page 12, discussion: the authors hypothesize that pre-LSC (in their model) arise from a committed progenitor with HSC properties and not from a true HSC, analogous to proposals in AML (see publications from the Weissman group). However, although this is conceivable, the provided evidence is not robust enough to make this conclusion. Further experiments would be required.

Reviewer #1

In this paper the authors show that restricted cell cycle in pre-LSCs is essential for clonal evolution and therapy resistance. Using an elegant mouse model they follow pre-LSC in *Lmo2^{Tg}* mice that were backcrossed with an inducible H2B-GFP and show that DN3 that do not proliferate have a increase self-renewal capacity able to serially transplant compared the cycling DN3-GFP^{lo} cells. They also show that these low cycling DN3 pre-LSC are more resistant to chemotherapy and are responsible for clonal evolution. There is nevertheless some concerns that will need to be addressed:

-Fig 2: The fact that non-cycling cells are engrafted better has been well-documented in HSC. For the DN3, it will be important that the decrease in engraftment is not due to lack or decrease in homing.

Response: Increased cycling leading to impaired homing of GFP^{lo} DN3 cannot explain the impaired engraftment because DN3 cells lacking p21 had increased engraftment (Figure 5E) despite increased cycling. We have added the following statement to the discussion.

“The impaired repopulating activity of GFP^{lo} DN3 thymocytes might be explained by increased cycling leading to impaired homing. However, this is highly unlikely as DN3 cells lacking p21 had increased repopulating activity in primary transplants (Fig. 5e) despite increased cycling.”

Could the authors explained why they used a 4 week setting for their serial transplant?

Response: As shown in our previous paper (McCormack et al. Science 2010), complete turnover of the thymus from HSCs takes approximately 3 weeks. Thus, we chose 4 weeks for serial transplants to allow any possible turnover from normal HSCs, which could be detected by the CD45.1 congenic marker. The following statement explaining the 4 week interval has been added to the results section.

“A period of 4 weeks between transplants was chosen to allow competition with normal HSCs, which take up to 3 weeks to repopulate the thymus.”

- Fig 2e: It is unclear why the DN3-GFP^{lo} cells are not reacquiring GFP. They should divided as they show at least in first transplant their capacity to engraft. Is that means that a 6 weeks chase is too long and that most of these cells have lose GFP. Some early chasing time should be provided to show that it is not an artefact. Also looking at the cell cycle status of the GFP^{lo} and GFP^{hi} after transplant will be helpful.

Response: The experiment described in fig 2e was designed to determine if DN3-GFP^{lo} cells could return to a slow cycling state. Therefore, we examined the cells 2 weeks after stopping doxycycline. The fact that there were no GFP^{hi} cells in the GFP^{lo} transplanted animals proves that they are unable to generate slow cycling cells. We did not look immediately at the end of doxycycline to show labelling but there is no reason to think that these cells would not be GFP-labelled whilst on doxycycline because GFP will be incorporated into any cell that replicates the H2B histone.

- In Fig 2S it will be of interest to show the proportion of DN3 cells that are GFP^{hi} versus GFP^{lo} after chemotherapy, an increase of DN3 GFP^{hi} fraction compared to DN3 GFP^{lo} but only from 9 to 2 %. It will be more informative to show the frequency of DN3GFP^{hi} and GFP^{lo} before and after chemotherapy. Are GFP^{hi} really enriched after chemotherapy and is this increase significant.

Response: We now show the proportion of GFP^{hi} cells following chemotherapy and irradiation in Fig S2e and g. This increase is statistically significant ($p < 0.001$).

- They then show that GFP^{hi} DN3 have an increase in p21 and by using p21KO demonstrate that DN3 pre-LSC lose their engraftment potential. I suppose the same will happened if inducing the GFP^{hi} cells into cycle by cytokine stimulation like possibly in this case IL-2. This will be more clinically relevant.

Response: We agree that stimulation of thymocytes with IL-2 would induce cell cycle. However, IL-2 would also promote cell survival, thus limiting the efficacy of chemotherapy (Lacobelli E *et al.* J Immunol

1999; Moriggi R *et al.* Immunity 1999). The *p21*-deficient mice were used to identify the key regulator of cell cycle in pre-LSCs without inducing a pro-survival response.

Overall, it is an interesting paper that provide direct evidence of the potential role of dormant pre-LSCs.

Reviewer #2

Tremblay and colleagues analyze a novel mouse line that allows non-dividing cells to be tracked (based on H2B-GFP label retention) in a T-ALL model (*Lmo2* transgenic). In this mouse model, they classify a subset of DN3 cells that are non-dividing to be pre-leukemic stem cells, which show serial transplantation capacity as well as greater chemotherapy resistance. Cell cycle inhibitor p21 was more highly expressed in this cell population, and by crossing with p21-KO mice, the authors demonstrate that loss of p21 induces cell division and differentiation, limiting leukemic capacity. While this is an interesting and novel study, several aspects of the work need to be address before it should be published, as outlined below.

Major comments:

1. How do the authors explain the 35% G_0 cells in the GFP^{lo} population in Figure 1E? Are these also cell-cycle restricted pre-LSCs?

Response: The H2B-GFP transgenic model and Ki67/DAPI are very different assays of cell cycle. The H2B-GFP system measures the 'history' of cell cycle once doxycycline is withdrawn. Therefore, the GFP^{lo} population represents cells that have divided at least 8 times over the 2 week period. Ki67/DAPI is a snapshot of cell cycle. Thus, 35% of these cells are in the G_0 phase at that one time. The fact that these cells in G_0 have proliferated extensively (at least 8 times in 2 weeks) and have lost their self-renewal capacity demonstrates that Ki67/DAPI is a less useful measure of so-called 'quiescence'.

2. Figure 2 demonstrates that GFP^{hi} DN3 cells can be serially transplanted while GFP^{lo} cells cannot. However, it is unclear whether GFP^{hi} cells can still be observed in these recipients. If the serially-transplantable leukemia come from proliferative cells (even if originally derived from quiescent cells), it questions the conclusion of the manuscript.

Response: Figure 2e shows that GFP^{hi} cells can be observed in secondary recipients of GFP^{hi} but not GFP^{lo} cells. Given that leukemias only arise in mice transplanted with GFP^{hi} cells (Figure 2d), this definitively shows that the origin of leukemias are from cell cycle-restricted GFP^{hi} cells.

3. Figure 2F shows that 9% of GFP^{hi} DN3 cells can survive chemotherapy. To demonstrate this is a useful model of pre-LSCs/disease relapse, it would be important to show that these cells are not senescent and are still functional pre-LSCs (or LSCs) and can drive disease progression in host mice or recipients (and that post-treatment leukemia does not arise from the GFP^{lo} cells).

Response: As stated previously, the H2B-GFP labelling model reflects the history of the cell cycle and therefore, the GFP^{hi} cells observed 24 hours after chemotherapy cannot be due to chemotherapy-induced senescence and must reflect cells that have not divided more than 3 times in the preceding 2 weeks.

4. Figure 4C/D nicely summarizes the gain and loss of chromosomes in GFP^{lo} cells and *Notch* mutations in GFP^{hi} cells. Are these mutations also found following serial transplantation of GFP^{hi} cells (or in the final T-ALL)? This could help to understand disease progression.

Response: Yes, *Notch1* mutations are frequently found in T-ALL arising from these mice as previously reported (Tremblay CS *et al.* Leukemia 2016).

5. Related to the above, the authors describe *Lmo2*-expression as the "pre-leukemic" mutation and that accumulation of additional mutations leads to leukemia. However, they then describe additional genetic abnormalities in pre-LSCs. At what point do these cells become LSCs or leukemia?

Response: Pre-LSCs are defined functionally by their ability to differentiate into normal progeny and give rise to leukemia after a 'prolonged period' (Sykes SM *et al.* Exp Hematol. 2015 – Ref 4). This functional definition does not preclude the possibility of additional mutations.

6. Given the availability of small molecule p21 inhibitors and claimed therapeutic relevance of the findings, it would be important to know whether p21 inhibitors can replace genetic deletion of p21 to induce differentiation and/or prevent T-ALL.

Response: Currently, there are no inhibitors of p21 that are sufficiently specific or potent for *in vivo* studies, which are essential for measuring pre-LSC activity. For example, a 'sorafenib-like' small molecule UC2288 targets p21 transcription at 10 μ M (Liu R *et al.* Future Med Chem 2013), but this is neither specific nor potent enough for *in vivo* studies.

7. Leukemias (and associated LSCs) are heterogeneous group of cancers. The authors should therefore be very careful about drawing broad conclusions from their results derived from one T-ALL mouse model. For example, with the current data, it is inappropriate to claim that "we show that cell cycle restriction is a fundamental property of pre-LSCs". Similarly, the title should accurately reflect the results.

Response: We agree and therefore have added "in a murine model of T-ALL" to the title and changed the last sentence of the discussion to "We propose the H2B-GFP system will be a powerful *in vivo* model to determine if similar properties apply to other models of pre-LSCs and identify and test strategies to overcome quiescence of relapse-inducing cells."

8. Revisions to the Figures are necessary to (1) highlight the key information (e.g. is Figure 3 necessary?) and (2) easy to follow (e.g. this reviewer found the layout of Figure 5 confusing).

Response: We have simplified the layout of Fig. 5 for improving clarity of the data.

Minor comments:

9. Throughout: The comparisons of DN3 pre-LSCs to HSCs throughout the manuscript is confusing and should be removed (or more carefully described).

Response: Quiescence is best understood in HSCs, where the same H2B-GFP transgene has been applied to identify GFP^{hi} HSCs (Foudi A *et al.* Nat Biotech 2009 – Ref 13). Therefore, we believe that the comparison between GFP^{hi} DN3 cells and quiescent HSCs is appropriate and clearly described in the manuscript.

10. Intro: This reviewer is unsure whether the LSC concept is limited to "rare cell population".

Response: We agree that the frequency of pre-LSCs may vary according to the type of leukemia and therefore we have removed the word "rare" from the first line of the introduction.

11. Intro: Given the background to pre-LSCs in leukemias, it would be appropriate to also discuss recent observations of clonal hematopoiesis.

Response: We agree and therefore have added "clonal hematopoiesis of the elderly" in the last line of the first paragraph of the intro.

12. Intro: It would also be appropriate to introduce the literature on *Lmo2* in T-ALL.

Response: We have added the sentence "Aberrant expression of *Lmo2* through chromosomal translocation or a somatically acquired neomorphic promoter occurs in 50% of T-cell acute lymphoblastic leukemia (T-ALL)." to the introduction.

13. Figure 1B: Based on the negative population, flow cytometry voltages differ between experiments but the positive gate remains the same.

Response: The GFP^{hi} gate for the 8 week post-pulse was erroneously copied and pasted from the 4 week pulse. The GFP^{hi} gate for 8 weeks has been redrawn but does not alter the result.

14. Figure S2D clearly shows an increase in numbers of GFP^{hi} DN3 cells in older mice and those with

T-ALL. But it is not clear when the pulses of Dox were provided. Where do these cells come from? Are they generated by differentiation or through self-renewal of DN3 cells? Additionally, given the age-related changes in lymphoid production, it would be important to compare this with the WT setting.

Response: We apologise that the description of this data was unclear in the original manuscript. The following description has been added to Fig. S2 legend:

“Absolute numbers of GFP^{hi} DN3 cells in the thymus of 2-month, 6-month old and leukemic *H2B-GFP;Lmo2^{Tg}* mice following 2 weeks of chase after labelling with doxycycline. Mean ± SD, Student’s *t*-test; ***p*<0.01, ****p*<0.001.”

The data shown in Fig. 5f is age matched, and shows no significant reduction in WT DN3 cells at 12 months of age.

15. Figure 4B and/or legend needs to be corrected.

Response: The legend of Figure 4B has been corrected.

16. Figure S6 legends need to be corrected.

Response: The legend of Figure S6 has been corrected.

17. Figure 6D – quantification of the biological repeats of this assay should be plotted. It appears p21KO does not completely rescue the *Lmo2* phenotype *in vitro*.

Response: We have now included the biological replicates for this assay (new Figure 6e).

18. Figure 6E – images displayed are difficult to understand. These should be either enlarged/higher resolution or moved to a supplementary figure and the data summary displayed alone.

Response: To improve clarity, we have now displayed a representative immunofluorescence analysis for both symmetrical and asymmetrical cell division, and included a summary of the division pattern for each genotype analysed (new Figure 6f).

19. It may be more appropriate to let others describe this work as “elegant”, rather than be one of the authors own conclusions.

Response: We have removed “elegant” from the discussion.

Reviewer #3

Tremblay et al examine cell cycle kinetics of pre-LSC by using their previously published *Lmo2* transgenic T-ALL mouse model crossed with a doxycycline inducible H2B-GFP mouse and later with p21^{-/-} mice.

Their model allows examination of GFP expression in dividing cells. Using the previously determined pre-LSC population of CD4-CD8-CD25+CD44- T cells (DN3 cells) the authors show that in *Lmo2^{Tg}* mice the population of GFP^{hi} cells remains significantly higher than in control mice and that this population resides to a large part in the G₀ phase of the cell cycle, indicating that these cells are cell-cycle restricted. By serially transplanting *Lmo2^{Tg}* pre-LSC into irradiated CD45.1+ recipient mice, the authors can nicely show that the repopulating capacity is contained only in the expanding GFP^{hi} cell population which is due in part to increased differentiation of T cells in GFP^{lo} population. In addition, secondary, tertiary and quaternary recipients of GFP^{hi} DN3 cells developed T-ALL, indicating acquisition of additional genetic events in pre-LSC. The authors then combine gene expression profiling and functional assays to identify p21 as a crucial pre-LSC regulating gene.

The question of pre-LSC functional properties and how to target these cells for durable elimination of leukemia is an important and enduring question in the field. As such, the establishment of this model and the performed experiments significantly contribute to understanding how pre-LSC may differ from normal HSC. Gene expression analyses and WE sequencing additionally provide a valuable data platform for further studies.

The experiments are sophisticated and well executed and sufficiently described. The authors have

performed an extensive amount of experiments which are very nicely presented in the Figures. They attempt to tie together pre-LSC properties, p21, aneuploidy and *Notch* in their discussion of their data. To me, these are separate observations and while very valid however are not entirely convincingly shown to be completely dependent on or a consequence of one another.

Response: We agree that these are separate observations, although all dependent on cell-cycle restriction: the increased therapeutic resistance, clonal expansion and long-term self-renewal properties of GFP^{hi} cells, as well as the exclusive presence of *Notch1* mutations in these leukemogenic cells suggest that cell cycle-restriction is crucial for pre-LSCs properties. Accordingly, the absence of p21 is associated with increased proliferation, aneuploidy and differentiation, as well as decreased clonal selection, impaired long-term self-renewal and limited leukemia development, suggesting that enforced cell cycle significantly compromises pre-LSCs properties. In the aggregate, our data confirms that clonal selection, leukemia development characterized by the acquisition of *Notch1* mutations and therapeutic resistance are dependent on the cell cycle-restriction of pre-LSCs.

Minor Queries:

1. FIGURE 6b and supplementary Fig6a: differential gene expression between *Lmo2*^{T9} and *Lmo2*^{T9} p21^{-/-} are much better visualized i.e. more convincing in the supplementary figure, would perhaps suggest to include in main manuscript.

Response: We agree. The data has now been included in Figure 6.

2. p21 is crucial for function of pre-LSC and pre-LSC with GFP^{hi} properties acquire *Notch* mutations. The authors postulate that this is due to DNA repair mechanisms. Are *Notch* mutations also found in the *Lmo2*^{T9} p21^{-/-} mice? Did the authors specifically look for this? Is acquisition of *Notch* mutations only a function of cell cycle?

Response: This is a good suggestion as it strengthens the evidence that loss of p21 leads to reduced clonal selection. We have now examined the expression of Notch1 variants in our RNA-seq data and found *Notch1* mutations in 3 month old *Lmo2* cells but not *Lmo2*p21^{-/-} cells. Furthermore, targeted sequencing of the *Notch1* locus in DN3 thymocytes isolated from 6-month old mice demonstrated a 2-fold reduction of Notch mutations in cells lacking p21 (Fig. S5h). We have added the following statement in the results and new data as Figure S5h.

“We previously showed that *Notch1* mutations are acquired during disease progression²⁰. To assess the importance of cell cycle in the acquisition of *Notch1* mutations, we used the RNA-seq data to identify variants differentially expressed in p21-deficient *Lmo2*^{T9} DN3 cells, and found that *Notch1* mutations were only present in *Lmo2*^{T9} thymocytes (Dataset 2). Targeted sequencing of the *Notch1* locus in DN3 thymocytes isolated from 6-month old mice revealed that the presence *Notch1* mutations was decreased by 2-fold in pre-LSCs lacking p21 (Fig. S5h).”

3. Chromosome additions are acquired in GFP^{lo} populations and thus postulated to be dependent on cell cycle. The authors speculate on mechanisms but provide no evidence. Again, do the *Lmo2*^{T9} p21^{-/-} mice exhibit aneuploidy?

Response: As presented in Fig. S5d, p21-deficient *Lmo2*^{T9} DN3 cells display increased aneuploidy as compared to *Lmo2*^{T9} DN3 thymocytes, confirming that increased cell cycle is associated with increased aneuploidy in pre-LSCs.

3. Page 12, discussion: the authors hypothesize that pre-LSC (in their model) arise from a committed progenitor with HSC properties and not from a true HSC, analogous to proposals in AML (see publications from the Weissman group). However, although this is conceivable, the provided evidence is not robust enough to make this conclusion. Further experiments would be required.

Response: Our previous work using a YFP reporter showed definitely that the pre-LSCs in this model arise from committed DN3 thymocytes and not HSCs (McCormack *et al.* Science 2010).

REVIEWERS' COMMENTS:

Reviewer #1 (Remarks to the Author):

The authors have provided adequate answer and extra data to address reviewers concerns. This new revised version is in my view highly improved.

Reviewer #2 (Remarks to the Author):

Tremblay and colleagues have answered the bulk of this Reviewer's concerns. The only outstanding concern is comment 3. Given the emphasis the authors place on developing a model to study pre-LSCs in relapse (see introduction), it is surely interesting to know whether the GFPhi cells can still induce leukemia post-chemotherapy, i.e. is the GFPhi population relevant in a chemotherapy/relapse model? This Reviewer was hoping for further analysis (than the 24-hour timepoint displayed) of the GFPhi cells post-chemotherapy to confirm that this population still the source of disease initiation post-chemotherapy. While further experiments may not be appropriate at this stage, it seems to be an important caveat to note.

Reviewer #3 (Remarks to the Author):

The authors have revised their manuscript according to the reviewers' suggestions. They have added new RNAseq analyses to confirm that NOTCH mutations are not found in p21 deficient mice which overall strengthens their data. The figures were updated for clarity.

Minor notes:

Most of Reviewer #2 comments were answered but not incorporated into the manuscript. Comments 1-3 would perhaps warrant incorporation into the text.

Same is true for reviewer #3 comment 3: if the authors previously showed that pre-LSC definitely arise from committed progenitors and not from HSC (cited Science paper 2010) then why are they now concluding this as a proposal from their current data? This argument does not makes sense to me.

Not all readers will be familiar with previous work from 2010- I suggest some additional clarification here or rewording.

New author F.Y. is not mentioned under Author contributions.

Reviewer #1

The authors have provided adequate answer and extra data to address reviewers concerns. This new revised version is in my view highly improved.

Reviewer #2

Tremblay and colleagues have answered the bulk of this Reviewer's concerns. The only outstanding concern is comment 3. Given the emphasis the authors place on developing a model to study pre-LSCs in relapse (see introduction), it is surely interesting to know whether the GFP^{hi} cells can still induce leukemia post chemotherapy, i.e. is the GFP^{hi} population relevant in a chemotherapy/relapse model? This Reviewer was hoping for further analysis (than the 24-hour timepoint displayed) of the GFP^{hi} cells post-chemotherapy to confirm that this population still the source of disease initiation post-chemotherapy. While further experiments may not be appropriate at this stage, it seems to be an important caveat to note.

Response: We agree this is an important question that remains to be directly addressed. Therefore, we have noted this in the discussion with the following sentence: "We did not directly examine the leukemic potential of the GFP^{hi} cells enriched by chemotherapy, however Ebinger et al. recently showed that label-retaining cells surviving chemotherapy in human B-ALL xenografts retained leukemia-initiating potential."

Reviewer #3

The authors have revised their manuscript according to the reviewers' suggestions. They have added new RNAseq analyses to confirm that NOTCH mutations are not found in p21 deficient mice which overall strengthens their data. The figures were updated for clarity.

Minor notes:

Most of Reviewer #2 comments were answered but not incorporated into the manuscript. Comments 1-3 would perhaps warrant incorporation into the text.

Response: We agree and therefore have added

- 'analysing snapshots of cell cycle' in the results section describing the data shown in Fig. 1d;
- 'Given that leukemias only arise in recipients injected with GFP^{hi} cells, our results demonstrate that' following the description of data shown in Fig. 2d;
- 'Given that the H2B-GFP labelling model reflects the history of the cell cycle, the enrichment for GFP^{hi} cells observed cannot be due to therapy-induced senescence and must reflect cells that have not actively divided in the preceding 2 weeks of chase.' prior to the last sentence summarizing the data from Fig. 2.

Same is true for reviewer #3 comment 3: if the authors previously showed that pre-LSC definitely arise from committed progenitors and not from HSC (cited Science paper 2010) then why are they now concluding this as a proposal from their current data? This argument does not makes sense to me.

Response: Our previous work showed that Lmo2 induces 'stem cell-like' self-renewal in a committed T-cell progenitor. This new work suggests that unlike normal HSCs, once these pre-LSCs become proliferative, they are unable to revert back to a restricted cell cycle.

Not all readers will be familiar with previous work from 2010- I suggest some additional clarification here or rewording.

Response: We agree. To improve clarity, we have rephrased this section of the introduction as follows "Transplant studies showed that pre-LSCs arise from immature CD4⁺CD8⁻CD25⁺CD44⁻ (DN3) T-cell

progenitors in *Lmo2* transgenic mice (*Lmo2*^{Tg}), as these cells were capable of long-term repopulation capacity in recipient mice. These self-renewing DN3 cells retained T-cell differentiation potential but eventually gave rise to T-ALL as they accumulated additional lesions that promote leukemia progression^{20, 21}. Importantly, these pre-LSCs could survive and recover after high dose radiation.”

New author F.Y. is not mentioned under Author contributions.

Response: Our apologies. This information has been added to the Authors contributions section.